# RL$^2$EAK: Reinforcement Learning Enhanced Prompt Leakage Attack in Multi-tenant Large Language Model Services

## Abstract

Large Language Models (LLMs) have become a transformative technology in both academia and industry. In practice, LLM services are typically deployed using multi-tenant serving frameworks. Popular inference frameworks such as vLLM and SGLang both employ Key-Value cache sharing among users to enhance computational efficiency. However, this shared caching mechanism may lead to potential leakage of the private user prompts. Previous works have demonstrated the impact of this information. Nevertheless, these works mainly focus on expanding the attack surface brought by the cache sharing mechanism, rather than optimizing the attack performance. This prevents users from accurately assessing the leakage's impact, thus hindering the timely leakage mitigation. To investigate the bounds of the cache-based side channel attack, we propose RL$^2$EAK, a reinforcement learning enhanced prompt leakage attack framework. We show that the adversary requires far fewer active prompt guesses with RL$^2$EAK than reported by previous works. To validate the effectiveness of our RL$^2$EAK, we apply RL$^2$EAK to two real-world scenarios, i.e., medical and finance, achieving a maximum $12.48\times$ reduction in average requests needed to guess one token. This study highlights the necessity for enhanced leakage transparency and careful management of cache-based information sharing, providing critical insights and references for future security countermeasures.

## 1 Introduction

Large Language Models (LLMs) have been widely deployed across various domains, e.g., Chatbots (OpenAI, 2025b; Claude, 2025) and GUI agents (Microsoft, 2025a). To improve inference efficiency and reduce computational costs, modern LLM service frameworks (e.g., vLLM (Kwon et al., 2023), SGLang (Zheng et al., 2023)) share Key-Value (KV) cache entries across user requests. Recently, both academia and industry have begun exploring persistent KV-cache storage to enhance cache reusability and reduce memory costs. For instance, (Lee et al., 2024; Sun et al., 2024) propose offloading KV-cache from GPU to CPU memory to alleviate expensive GPU memory usage. Similarly, vLLM integrates such KV-cache management approaches into its framework(vLLM, 2025). As storage capacity becomes increasingly affordable and cache persistence more prevalent, it becomes urgent to understand and manage the privacy risks of cached queries.

Prior research has shown that cache-sharing strategies in LLM services can introduce different types of side channels, such as Time to First Token (TTFT) (Zheng et al., 2024; Song et al., 2024) and ordering based on LLM scheduling policies (Wu et al., 2025). These side channels enable adversaries to craft malicious requests for revealing matches with other users and recover sensitive user query information. However, these studies primarily focused on exploring the expanded attack surface within cache sharing mechanisms, overlooking the practical efficiency of side-channel exploitation. Prior works (Gu et al., 2025; Luo et al., 2025) indicate that extracting information typically requires an impractically large number of requests. This inefficiency stems from not fully accounting for a real-world adversary's capability to optimize the attack strategy. Consequently, the true severity of these privacy risks remains underestimated. This gap hinders accurate assessment of leakage severity and the development of countermeasures.

To fill this gap, we investigate how to maximize real adversary attack capabilities through systematic model optimization in this paper. We consider a practical threat model where the adversary is assumed to have knowledge of the general domain of user queries. This information can be inferred through various means, such as application context, service endpoints, or metadata. This assumption about the adversary's knowledge is realistic, especially given the increasing number of organizations that deploy private LLMs or utilize private LLM inference services from third-party cloud providers (JP-Morgan, 2024; Dataprocorp, 2025; Dennstädt et al., 2025). Users of these domain-specific LLM services typically inquire about topics within specific domains, e.g., financial analysis, medical diagnosis, or legal research. Consequently, attackers can leverage public datasets from similar domains to finetune their local models, allowing for efficient side-channel attacks.

Training in a domain-aware setting presents two main challenges. First, Supervised Fine-Tuning (SFT) tends to overfit and struggles to effectively learn domain-specific knowledge since specific-domain QA datasets are still dominated by general language, which makes it difficult for the model to grasp specialized terminology. Second, Reinforcement Learning (RL) algorithms, e.g., Proximal Policy Optimization (PPO) (Schulman et al., 2017) and Group Relative Policy Optimization (GRPO) (Shao et al., 2024), face the complexity of reward engineering. Rewards based on semantic similarity can lead to overfitting on superficial patterns, while rewards that rely on exact token matching tend to provide insufficient feedback.

To enable preference alignment for domain-specific knowledge while overcoming the challenging reward design, we propose $RL^2EAK$, a two-stage fine-tuning framework with a novel automated annotation approach. In the first stage, we apply SFT on the base local model using domain-specific data to familiarize it with relevant knowledge in that specific field. In the second stage, we leverage this SFT-tuned model to automatically annotate the training data for DPO. This automatic annotation involves identifying what we refer to as "hard tokens", which are domain-specific terms that are challenging to generate but contain crucial and sensitive domain information. In this way, $RL^2EAK$ eliminates the need for manual annotation and prioritizes the extraction of genuinely sensitive, domain-specific information, guided by feedback on the identified hard tokens.

Through comprehensive evaluation, we show that $RL^2EAK$ achieves significant performance improvements across two domain-specific datasets: MedQA (Jin et al., 2021) and FinanceBench (Islam et al., 2023). Compared to the baseline approach that uses base models as local LLMs for query reconstruction (Wu et al., 2025), $RL^2EAK$ achieves a $12.48\times$ reduction in Average Token Requests Per Query (APR) on FinanceBench with Qwen2.5-3B-Instruct (Yang et al., 2024) as the backbone and demonstrates substantial performance gains on MedQA. Furthermore, we analyze the impact of SFT overfitting on adversarial performance, which further validates $RL^2EAK$'s effectiveness.

In summary, our contributions are as follows:

- We identify that previous works primarily focus on expanding the attack surface introduced by cache-based mechanisms, rather than optimizing attack effectiveness. We first attempt to maximize the adversary's capabilities through systematic model optimization.
- We design an RL-enhanced attack framework that leverages SFT and DPO to improve the efficiency of side-channel attacks. We further develop an automated annotation method for DPO to avoid human labeling.
- We conduct comprehensive evaluations across two knowledge-intensive domains (medical and finance) to demonstrate $RL^2EAK$'s effectiveness and provide quantitative analysis of performance improvements.
- We provide extensive discussion of $RL^2EAK$'s compatibility with other active side channels, and potential application as a side channel risk assessment tool.

## 2 RELATED WORKS

In this section, we describe the background of $RL^2EAK$, including side channel attacks in LLM services and RL-based red teaming for LLMs.

**Side Channel Attacks in LLM Services.** Current side channel attacks in LLM services can be divided into two types: passive side channel attacks (Weiss et al., 2024; Zhang et al., 2024a; Wei et al., 2024) and active side channel attacks (Gao et al., 2025; Song et al., 2024; Wu et al., 2025;

Zheng et al., 2024; Adiletta & Sunar, 2025; Luo et al., 2025; Zhang et al., 2024b). In a passive side channel attack, the adversary first establishes fingerprints via passively monitoring the queries between the user and the victim LLM or by interacting with public LLMs, and then utilizes these fingerprints to identify user queries or intents. For instance, Weiss et al. (2024) infer the character length of each token in LLM responses by analyzing encrypted network traffic between the user and LLM, thereby revealing the user's prompts. Zhang et al. (2024a) identifies user queries by analyzing timing patterns in LLM response generation. Furthermore, Wei et al. (2024) and Soleimani et al. (2025) exploit speculative decoding mechanisms to build fingerprints for different user queries.

In an active side channel attack, the attacker actively interacts with or manipulates the victim LLM or its underlying system (e.g., hardware caches or serving mechanisms) during the user's query process. For example, Adiletta & Sunar (2025) and Gao et al. (2025) demonstrate that attackers can infer user queries by monitoring hardware cache changes to determine the position of accessed embedding vectors. Zheng et al. (2024) leverages timing-based side channels by detecting whether queries hit the cache to determine if proposed dummy queries match users' actual queries. Wu et al. (2025) demonstrates that serving ordering mechanisms (e.g., longest prefix matching) also reveal side channel information when proposed queries match with targeted ones. Our work distinguishes itself from previous works by employing RL to enhance cache-based active side channel attacks. Rather than expanding attack scenarios, we focus on optimizing the probing query generation process to explore the maximum capacities of real-world adversaries.

**RL-Based Red-Teaming for LLMs.** RL-based red teaming has emerged as a prominent paradigm for discovering vulnerabilities of LLMs in security and privacy. A series of works (Paulus et al., 2024; Chen et al., 2024; Tang et al., 2024) leverage RL for enhancing jailbreaking techniques, enabling effective evasion of the victim LLM's guardrail and alignment to generate malicious content. Similarly, RL is employed to prompt LLMs to produce harmful responses that incorporate toxic content, sensitive terminology, or factual errors (Perez et al., 2022; Casper et al., 2023; Zheng et al., 2025; Zhao et al., 2025). Recently, RL is applied to privacy leakage attacks against LLMs. Nie et al. (2025) utilizes RL for system prompt extraction (Carlini et al., 2021; Li et al., 2022; Wang et al., 2023) and training data extraction attacks (Nasr et al., 2025) with word edit similarity as a reward function to elicit private information from the victim LLM. In contrast to previous research, our work target a distinct privacy attack scenario, leveraging RL to optimize adversarial strategies for recovering users' real-time prompts through cache-sharing side channels.

## 3 PROBLEM FORMULATION

In this section, we describe the attacking surface of the LLM serving system (i.e., side channel by cache sharing), our threat model, and the detailed attack scenario.

### 3.1 SIDE CHANNEL BY CACHE SHARING MECHANISM IN MULTI-TENANT LLM SERVICE.

Current multi-tenant LLM service frameworks (e.g., vLLM (Kwon et al., 2023), SGLang (Zheng et al., 2023), LightLLM (Nakandala et al., 2020)) are primarily based on Transformer (Vaswani et al., 2017) model architecture. One core component of these LLM models is the self-attention mechanism. In the inference process, each token $x_i$ is represented by three vectors through linear projections: query $q_i = W_q x_i$, key $k_i = W_k x_i$, and value $v_i = W_v x_i$, where $W_q$, $W_k$, and $W_v$ are trained weight matrices. The attention mechanism then computes the weighted combination of values based on the similarity between queries and keys, that is, $\mathrm{Attention}(Q, K, V) = \mathrm{Softmax}\left(QK^T/\sqrt{d_k}\right)V$, where $Q$, $K$, and $V$ represent the matrices of queries, keys, and values, respectively, and $d_k$ is the dimension of the key vectors.

During autoregressive text generation, the KV cache mechanism stores the computed key and value vectors for all previously processed tokens, eliminating the need for redundant calculations in subsequent decoding steps. Due to the causal nature of the attention mask in LLMs, the key and value representations for a given token depend only on the preceding tokens in the sequence. This property enables KV cache sharing across different requests: when multiple users submit prompts with identical prefixes, the cached key-value pairs for these shared prefix tokens can be reused, significantly reducing both memory consumption and computational overhead (Kwon et al., 2023; Zheng et al., 2023). The cache sharing architecture operates through a unified memory pool where requests

from different tenants can access shared cached states. When a cache hit occurs, the system directly utilizes stored values, thereby reducing TTFT and overall response latency. However, while this optimization accelerates inference performance, it concurrently introduces security vulnerabilities. The shared nature of these caches creates an observable timing side-channel, as cache hits result in substantially faster response times compared to cache misses, potentially leaking information about other users' inputs through timing analysis.

## 3.2 THREAT MODEL

In our threat model, the adversary's goal is to reconstruct prompts sent by other victim clients through cache-based side channels. Following the assumptions established by Wu et al. (2025) and Zheng et al. (2024), the adversary possesses capabilities equivalent to an ordinary LLM client, with only black-box access to the LLM server (i.e., the adversary has no privilege to access the model architecture or parameters). The LLM operates in streaming mode, delivering responses to users in real time. We assume the adversary can: (1) send queries (with a maximum limit for each victim query) to the LLM server, and correspondingly measure TTFT to detect cache hits; (2) access publicly available tokenizers, which are commonly provided by online LLM services such as OpenAI (2024) and open-source LLMs (Qwen, 2025). The adversary cannot view internal server logs, modify server configurations, or access other users' communication channels directly.

We further assume the adversary has domain knowledge about victim clients' queries (e.g., awareness that a user is a doctor who probably submits medical-related queries) but lacks knowledge on specific prompt templates or the exact query content used by the victim client (Wu et al., 2025; Soleimani et al., 2025). This assumption is practical, as more and more data-sensitive institutions (e.g., Medical (Dennstädt et al., 2025), Legal (Dataprocorp, 2025), Financial (JP-Morgan, 2024)) choose to privately deploy LLMs or lease private LLM services from third-party cloud providers. For example, in a hospital's LLM deployment, an adversary (such as a malicious staff member or a compromised account) can assume that most queries are medical-related, but still cannot access the specific content of other users' sensitive patient information.

## 3.3 ATTACK SCENARIO

Figure 1 illustrates our attack scenario in the context of cache-based prompt leakage attacks. Our system consists of two parties: victim clients and a cloud-based LLM inference server. The server concurrently receives queries from multiple users, with these requests being processed on shared GPU nodes. We assume that the victim's and attacker's queries are processed on a server node sharing the same KV cache pool. This is a realistic assumption, as many production deployments (e.g., Google Cloud. (2025), Microsoft (Feng et al., 2025), ByteDance Inc. (2025)) currently employ KV cache pooling to persist cached key-value pairs over extended periods. Compared to traditional physical GPU node allocation, the pooling architecture increases the likelihood that attackers and victims share the same KV cache node. Moreover, this assumption aligns with prior work on KV cache side-channel attacks (Wu et al., 2025; Zheng et al., 2024; Song et al., 2024).

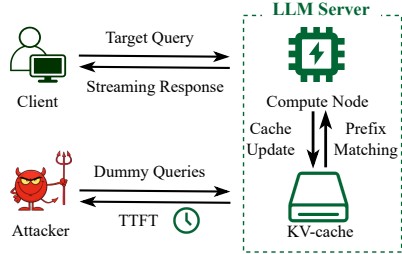

Figure 1: Attack scenario overview

At a high level, the attack process operates as follows: (1) The victim client sends the target query to the LLM server, and the query's tokens are stored in the KV-cache. (2) The adversary then sends multiple specifically-designed dummy queries to the LLM server, attempting to match prefixes of the target query. (3) The adversary observes each response's TTFT to determine whether a cache hit occurs, which indicates whether the queried tokens match the targeted ones. RL$^2$EAK operates iteratively, with each iteration recovering one token from the victim's prompt. The adversary terminates when either the entire targeted prompt is reconstructed or the maximum number of attempts is reached.

Figure 2: An overview of RL$^2$EAK's operation pipeline.

## 4 METHODOLOGY

We now introduce RL$^2$EAK, an automated prompt leakage attack framework that leverages Supervised Fine-Tuning (SFT) and Direct Preference Optimization (DPO) to fine-tune an adversarial model for probe query generation. We first describe our RL-based approach for automatically optimizing adversary capabilities and then present the order-based KV-cache side channel attack. Figure 2 provides an overview of RL$^2$EAK. The corresponding operational steps are detailed in Algorithm 1 within Appendix A.

### 4.1 LOCAL MODEL FINE-TUNING WITH REINFORCEMENT LEARNING

RL$^2$EAK involves a two-stage approach for local model fine-tuning in RL$^2$EAK, i.e., first SFT then DPO. We assume an auxiliary dataset for SFT based on the adversary's domain knowledge, as mentioned in Section 3.2. For DPO, we propose a novel automated annotation mechanism.

**Supervised Fine-Tuning.** Given an auxiliary dataset $\mathcal{D}_{\text{aux}} = \{(x, y)\}$, where $x$ and $y$ are user query and LLM response separately, we first apply SFT to the base adversarial model $\pi_{\text{base}}$ via maximizing the likelihood of the user query and the corresponding response:

$$\mathcal{L}_{\text{SFT}}(\theta) = \sum_{s \in \mathcal{D}_{\text{aux}}} \sum_{t=1}^{|s|} \log \pi_\theta(s_t | s_{<t}), \tag{1}$$

where $s = [x, y] = [s_{<t}, s_t]$ denotes the concatenated sequence of user query and LLM response, $|s|$ is the number of tokens in $s$.

**Direct Preference Optimization With Automated Annotation.** After SFT, we obtain $\pi_{\text{SFT}}$. Though extending training epochs of SFT is a traditional approach for performance improvement, we find in our experiment that increasing SFT training epochs leads to poor attack performance, primarily due to overfitting and mode collapse. As demonstrated in our ablation study (Figure 3), extending SFT training causes the number of guessing attempts to increase to $3.70\times$ compared with the baseline. To further improve the attacker's capacity, we leverage DPO for better alignment of the adversarial model with the domain-specific dataset. Our key insight stems from the observation that there is a small subset of queries that are challenging to be predicted. This motivates us to develop an automated mechanism to identify these challenging queries for better preference alignment, thereby enhancing overall model capacity.

The *automated annotation* approach we propose for DPO leverages token prediction difficulty (i.e., likelihood of each token) as a signal for constructing preference pairs. Our approach begins by identifying hard-to-recover tokens. Specifically, we use $\pi_{\text{SFT}}$ to predict each token $s_t$ in each sample $s = [x, y]$ of the auxiliary dataset $\mathcal{D}_{\text{aux}}$ and rank the target token in the descending order based on its likelihood $\pi_{\text{SFT}}(s_t | s_{<t})$. Intuitively, this ranking represents the recovery difficulty, where tokens with higher rankings indicate higher difficulty. Formally, we identify all preferred responses $s^{\text{win}}$ via

$$\{s^{\text{win}} = [s_{<t}^{\text{win}}, s_t^{\text{win}}] \mid [s_{<t}, s_t] \, \forall s \in \mathcal{D}_{\text{aux}} \text{ if } \text{Rank}(s_t) > \gamma\}, \tag{2}$$

where $\mathrm{Rank}(s_t)$ is a function that returns the likelihood ranking of $s_t$ among the likelihood set $\{\pi_{\mathrm{SFT}}(v|s_{<t})|v \in \mathcal{V}\}$ ($\mathcal{V}$ is the vocabulary). Then, for each prefered resposne $s^{\mathrm{win}} = [s^{\mathrm{win}}_{<t}, s^{\mathrm{win}}_t]$, we construct the corresponding dispreferred response $s^{\mathrm{lose}}$ using $\pi_{\mathrm{SFT}}$ via greedy decoding at the position of the hard token $s_t$:

$$s^{\mathrm{lose}} = [s^{\mathrm{lose}}_{<t}, s^{\mathrm{lose}}_t] = [s^{\mathrm{win}}_{<t}, \arg\max_{v \in \mathcal{V}} \pi_{\mathrm{SFT}}(v|s^{\mathrm{win}}_{<t})], \tag{3}$$

That is, $s^{\mathrm{lose}}_{<t} = s^{\mathrm{win}}_{<t}$ and $s^{\mathrm{lose}}_t = \arg\max_{v \in \mathcal{V}} \pi_{\mathrm{SFT}}(v|s^{\mathrm{win}}_{<t})$ (the greedy decoding operator). Through this, we create pairs $(s^{\mathrm{win}}, s^{\mathrm{lose}})$ where each $s^{\mathrm{win}}$ contains a hard token $s^{\mathrm{win}}_t$, while the corresponding $s^{\mathrm{lose}}$ contains a highly confident yet incorrect token at the same position. To clarify this, we provide an example in Figure 2. Given a samlpe $s$ "Patient presents skin rash" in the dataset, we identify "skin" is a hard token since its likelihood ranking is the 478-th (exceeding our threshold) and thus form a preference pair: prefix "Patient presents" as $s^{\mathrm{win}}_{<2}$, the preferred token "skin" as $s^{\mathrm{win}}_2$, and the dispreferred token $s^{\mathrm{lose}}_2$ sampled from SFT-tuned model.

We then construct the preference dataset $\mathcal{D}_{\mathrm{pref}} = \{(s^{\mathrm{win}}, s^{\mathrm{lose}})\}$ and apply DPO (Rafailov et al., 2023) using $\mathcal{D}_{\mathrm{pref}}$ to improve $\pi_{\mathrm{SFT}}$. The loss function for DPO is thus:

$$\mathcal{L}_{\mathrm{DPO}}(\theta) = -\mathbb{E}_{(s^{\mathrm{win}}, s^{\mathrm{lose}})} \sim \mathcal{D}_{\mathrm{pref}} \left[ \log \sigma \left( \beta \log \frac{\pi_\theta(s^{\mathrm{win}}_t|s^{\mathrm{win}}_{<t})}{\pi_{\mathrm{ref}}(s^{\mathrm{win}}_t|s^{\mathrm{win}}_{<t})} - \beta \log \frac{\pi_\theta(s^{\mathrm{lose}}_t|s^{\mathrm{lose}}_{<t})}{\pi_{\mathrm{ref}}(s^{\mathrm{lose}}_t|s^{\mathrm{lose}}_{<t})} \right) \right], \tag{4}$$

where $\pi_{\mathrm{ref}}$ is the reference model in DPO that is initialized from $\pi_{\mathrm{SFT}}$ and remains fixed during training, $\beta$ is a hyperparameter that controls the strength of the KL regularization, and $\sigma$ is the sigmoid function. This approach maximizes the likelihood of each preferred response $s^{\mathrm{win}}$ relative to its corresponding dispreferred responses $s^{\mathrm{lose}}$ while maintaining the model's overall capabilities through KL regularization with respect to $\pi_{\mathrm{ref}}$.

## 4.2 Order-based KV-cache Prompt Leakage Attack

Let the victim user's targeted query we want to recover be $s^{\mathrm{victim}}$, which contains $n$ tokens. To launch a specific prompt leakage attack, we leverage the Longest Prefix Match (LPM) scheduling policy utilized by SGLang (Zheng et al., 2023), one of the most prominent LLM inference frameworks. This policy ensures that when the request queue contains multiple queries, waiting requests are prioritized based on the length of their matched prefix tokens. To guess a token $s^{\mathrm{victim}}_t$, we send a batch of queries $Q = \{\tilde{q}^1, ..., \tilde{q}^k\}$ to the LLM server, where $k = 2m + |Q_{\mathrm{gen}}|$ represents the total number of queries. The batch contains $Q_{\mathrm{gen}}$ candidate queries (our generated guesses for position $i$) placed between two groups of $m$ dummy queries each. Each dummy query shares the same low-probability token obtained from local LLM predictions. All generated queries share the same prefix $q_{<t}$ and differ only in the last token $q_t$. After sending the entire batch, we observe the response order with TTFT to determine if a cache hit occurs. Specifically, if a cache hit occurs, the query containing the correct token match will be prioritized due to LPM scheduling, causing a gap between consecutive real queries in the response sequence. To detect cache hit $\mathsf{D}_{\mathrm{hit}}$, we use:

$$\mathsf{D}_{\mathrm{hit}} = \begin{cases} 1, & \exists \tilde{q}^i \in Q_{\mathrm{gen}} \text{ s.t. } \left[ \min_{\tilde{q} \in Q_{\mathrm{gen}} \setminus \{\tilde{q}^i\}} \mathrm{pos}(\tilde{q}) \right] - \mathrm{pos}(\tilde{q}^i) > \lceil \theta \cdot m \rceil \\ 0, & \text{otherwise,} \end{cases} \tag{5}$$

where $\mathrm{pos}(\tilde{q})$ is the response position of query $\tilde{q}$, $\theta \in (0, 1)$ is a parameter to adjust the sensitivity of cache hit detection. When $\mathsf{D}_{\mathrm{hit}} = 1$, we identify the query $\tilde{q}^i$ contains the correct token.

# 5 Experiment

## 5.1 Experimental Setup

We implement and evaluate RL$^2$EAK on a Linux-based operating system, and more detailed experiment settings are described below. For our implementation, we leverage SGLang (Zheng et al., 2023) as the LLM serving and inference framework.

**Baselines & Benchmarks.** We conduct evaluation using three specific LLMs from the Qwen-2.5 series (Yang et al., 2024) as baseline local LLMs: Qwen-2.5-3B-instruct, Qwen-2.5-7B-instruct,

Table 1: Main experimental results on MedQA, FinanceBench, and PubMedQA datasets. DPO results are obtained by directly training the base model using RL$^2$EAK's auto-annotation approach.

| Model | Method | MedQA | | | | | | FinanceBench | | | | | | PubMedQA | | | | | |
|---|---|---|---|---|---|---|---|---|---|---|---|---|---|---|---|---|---|---|---|
| | | ASR$_{500}$ | ASR$_{1000}$ | ASR$_{10000}$ | w/l | ATRT | APR | ASR$_{500}$ | ASR$_{1000}$ | ASR$_{10000}$ | w/l | ATRT | APR | ASR$_{500}$ | ASR$_{1000}$ | ASR$_{10000}$ | w/l | ATRT | APR |
| Qwen-2.5-3B-Instruct | Base | 18.0% | 37.3% | 95.3% | - | 2.59 | 41.99 | 2.0% | 2.0% | 54.0% | - | 9.92 | 240.81 | 0.0% | 3.0% | 68.0% | - | 22.06 | 628.50 |
| | SFT | 27.3% | 53.3% | 98.0% | 80.0% | 2.20 | 26.37 | 88.0% | 90.0% | 100.0% | 100.0% | 1.61 | 20.81 | 12.0% | 28.0% | 94.0% | 91.0% | 8.39 | 220.85 |
| | DPO | 18.0% | 38.0% | 95.3% | 48.7% | 2.58 | 41.86 | 2.0% | 2.0% | 56.0% | 68.0% | 9.90 | 238.81 | 1.0% | 5.0% | 71.0% | 63.0% | 19.70 | 571.51 |
| | RL$^2$EAK | 29.3% | 55.3% | 98.6% | 84.0% | 2.12 | 21.60 | 88.0% | 90.0% | 100.0% | 100.0% | 1.59 | 19.29 | 11.0% | 31.0% | 94.0% | 91.0% | 7.68 | 208.72 |
| Qwen-2.5-7B-Instruct | Base | 18.0% | 36.0% | 94.0% | - | 2.71 | 54.50 | 6.0% | 6.0% | 66.0% | - | 9.87 | 259.65 | 0.0% | 1.0% | 49.0% | - | 31.66 | 943.05 |
| | SFT | 24.0% | 49.3% | 94.7% | 79.3% | 2.47 | 44.50 | 76.0% | 82.0% | 96.0% | 94.0% | 2.32 | 69.63 | 9.0% | 21.0% | 86.0% | 91.0% | 10.93 | 291.28 |
| | DPO | 17.3% | 37.3% | 94.7% | 55.3% | 2.68 | 52.67 | 6.0% | 6.0% | 72.0% | 84.0% | 8.94 | 246.38 | 0.0% | 1.0% | 49.0% | 70.0% | 30.24 | 901.09 |
| | RL$^2$EAK | 22.7% | 48.0% | 96.0% | 81.3% | 2.35 | 40.49 | 86.0% | 92.0% | 96.0% | 96.0% | 2.19 | 66.87 | 6.0% | 22.0% | 88.0% | 91.0% | 10.56 | 274.91 |
| Qwen-2.5-14B-Instruct | Base | 23.3% | 45.3% | 94.7% | - | 2.47 | 44.35 | 4.0% | 4.0% | 64.0% | - | 8.49 | 214.19 | 0.0% | 0.0% | 58.0% | - | 28.20 | 796.08 |
| | SFT | 31.3% | 56.6% | 96.0% | 78.6% | 2.14 | 33.69 | 78.0% | 88.0% | 98.0% | 96.0% | 1.98 | 39.49 | 0.0% | 24.0% | 81.0% | 88.0% | 12.17 | 374.85 |
| | DPO | 24.0% | 46.0% | 94.7% | 68.0% | 2.45 | 43.83 | 4.0% | 4.0% | 64.0% | 62.0% | 8.16 | 213.46 | 0.0% | 0.0% | 59.0% | 40.0% | 25.37 | 753.28 |
| | RL$^2$EAK | 30.7% | 60.0% | 97.3% | 82.0% | 2.07 | 30.60 | 84.0% | 88.0% | 98.0% | 98.0% | 1.85 | 36.31 | 12.0% | 30.0% | 89.0% | 88.0% | 11.21 | 304.76 |
| Llama-3.1-8B-Instruct | Base | 5.3% | 27.3% | 92.0% | - | 3.16 | 60.00 | 0.0% | 0.0% | 60.0% | - | 10.96 | 286.52 | 0.0% | 0.0% | 57.0% | - | 27.81 | 801.71 |
| | SFT | 8.7% | 43.3% | 98.0% | 74.0% | 2.35 | 30.43 | 62.0% | 86.0% | 98.0% | 98.0% | 2.04 | 37.91 | 13.0% | 33.0% | 93.0% | 94.0% | 8.81 | 254.86 |
| | DPO | 8.0% | 28.7% | 92.7% | 74.0% | 3.02 | 54.86 | 0.0% | 2.0% | 66.0% | 88.0% | 9.61 | 248.69 | 0.0% | 0.0% | 60.0% | 84.0% | 22.00 | 666.01 |
| | RL$^2$EAK | 12.6% | 48.7% | 98.7% | 79.3% | 2.17 | 28.31 | 64.0% | 86.0% | 100.0% | 98.0% | 1.98 | 33.89 | 14.0% | 33.0% | 93.0% | 94.0% | 8.47 | 241.48 |

Qwen-2.5-14B-instruct, and Llama-3.1-8B-instruct (Team, 2024). We use three real-world domain-specific datasets in the evaluation: MedQA (Jin et al., 2021) and PubMedQA (Jin et al., 2019) in the medical domain and FinanceBench (Islam et al., 2023) in the finance domain. For MedQA, we utilize the English subset comprising 10,178 training samples, 1,272 validation samples, and 1,273 test samples. Considering the computational cost of launching prompt leakage attacks, we randomly sample 150 instances from the test set for evaluation, which is consistent with the experimental setup in prior prompt leakage attack studies (Wu et al., 2025; Zheng et al., 2024; Song et al., 2024). For FinanceBench, which lacks a predefined train-test split, we perform a random stratified split of the complete dataset, allocating 400 samples for training and 50 samples each for validation and testing to ensure sufficient evaluation coverage. For PubMedQA (Jin et al., 2019), we utilize the 1k expert-labeled subset of medical QA data. Following the same evaluation protocol as the other two benchmarks, we allocate 800 samples for training and 100 samples each for validation and testing.

**Evaluation Metrics.** In our evaluation, we employ three metrics:

- *Attack Success Rate* (ASR) measures the success rate of prompt leakage attacks. To more precisely represent the adversary's capacity and account for the characteristics of real-world KV-cache systems, we adopt different maximum request limits. For instance, we set a maximum limit of 10,000 request attempts, denoted as ASR$_{10000}$.

- *Average Number of Requests per Token* (APR) tracks the average number of requests required to recover a single token. This metric provides an overall assessment of the impact of model optimization methods at the token level.

- *Request Number Win/Lose Rate* (w/l) measures the proportion of queries where the optimized model requires fewer requests compared to the base model. It directly quantifies the relative effectiveness between the base model and the optimized model.

- *Average Token Recovery Time* (ATRT) measures the average time required to recover a single token. This metric provides a direct and intuitive assessment of the practical improvement in efficiency when recovering user queries.

## 5.2 MAIN RESULTS

In our main experiments, we compare RL$^2$EAK with the base LLM prediction approach (Wu et al., 2025) and SFT/DPO models on both benchmarks. We use "Help me to guess the input:" as the consistent prompt prefix for all prompt leakage methods. The prefix serves two purposes: instructing the LLM to perform the query reconstruction task and bootstrapping the auto-regressive generation process. We set both $m$ and $Q_{gen}$ to 20, aligning with the configuration used in the baseline (Wu et al., 2025). To ensure a fair comparison, Qwen-2.5-3B-Instruct was employed as the server model across all experiments." Table 1 presents the main results. Overall, RL$^2$EAK substantially enhances the local LLM's performance for prompt leakage attacks across all evaluated models and benchmarks, demonstrating consistent performance improvements as a prompt leakage adversary. Furthermore, the experimental results reveal several important findings:

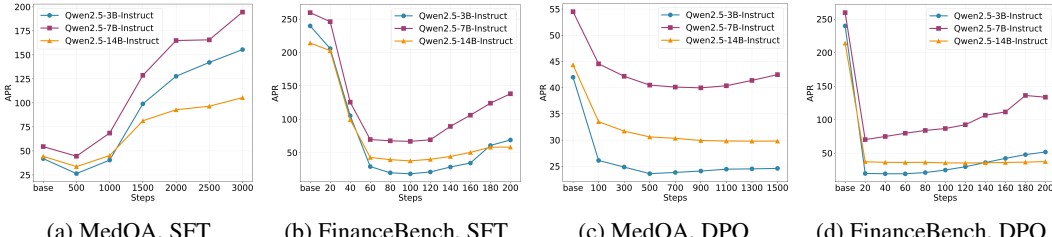

(a) MedQA, SFT    (b) FinanceBench, SFT    (c) MedQA, DPO    (d) FinanceBench, DPO

Figure 3: Ablation Study Results. The left figures show APR changes during SFT training, while the right figures show DPO training results based on SFT-tuned models (MedQA: SFT training step 500, FinanceBench: SFT training step 100).

**SFT Significantly Enhances Adversarial Capability and Serves as a Critical Component.** Our results demonstrate that SFT consistently improves the adversary's capacity to launch prompt leakage attacks. Compared to base LLMs, models enhanced with SFT show a 16.0% $ASR_{1000}$ improvement on MedQA when using Qwen-2.5-3B-Instruct, along with a 37.2% APR reduction. The improvements are more significant in FinanceBench, where the domain-specific language patterns are more consistent. The consistent improvement trends observed in ATRT further validate this finding. The explanation for this is that SFT enables the model to learn the general patterns and linguistic characteristics of specific domains, thereby developing a better understanding of how prompts are typically structured within those specific domains.

Additionally, the comparison between DPO and the $RL^2EAK$ demonstrates the necessity of the SFT process. With only DPO applied, the adversary's attack performance remains limited; while it can achieve an effective win/loss rate over 50.0%, there is no significant improvement in the APR. This lack of progress occurs because DPO alone does not enhance the adversary's ability to predict "hard tokens," which consume substantial request numbers, indicating that DPO struggles to yield significant improvements for side-channel attacks. However, when DPO is applied after SFT, it can substantially enhance the SFT model. Specifically, on the MedQA benchmark, $RL^2EAK$ reduces the APR from 26.37 to 21.60, representing a relative reduction of 18.1% compared to the SFT baseline. Similar relative reductions of approximately 10% are observed across other settings. This consistent improvement is particularly significant because, as the APR decreases, achieving further reductions while maintaining model stability becomes increasingly challenging. $RL^2EAK$ achieves the lowest APR across all settings and benchmarks, validating the effectiveness of the SFT-then-DPO approach.

**Model Scale Does Not Guarantee Better Attack Performance.** Contrary to conventional expectations, our experiments illustrate that larger models do not consistently outperform their smaller versions in prompt leakage tasks. In both datasets' evaluations, small models achieved comparable or even superior attack success rates compared to large ones. Notably, Qwen-2.5-3B-Instruct consistently achieved the lowest APR and the highest ASR across both benchmarks, despite not having the best base capabilities among the tested models. This phenomenon can be attributed to the characteristic of prompt leakage tasks, where targeted optimization and fine-tuning alignment are more critical than model size. Consequently, well-trained smaller models may outperform large-scale models that lack specialized optimization for adversarial objectives. These findings suggest that researchers should prioritize model architecture and training strategies over model scale when developing prompt leakage attacks.

### 5.3 ABLATION STUDY ON DIFFERENT TRAINING EPOCHS

We conduct an ablation study on training parameters across both benchmarks, with results illustrated in Figure 3. The left two figures display the changes in APR during the SFT stage. The results reveal that all models initially experience a reduction in APR during the early training steps, followed by a gradual increase over time. Notably, the dataset MedQA appears to be more susceptible to severe overfitting as training steps increase, with the APR rising to 3.70× that of the base model after 3000 training steps. In contrast, the DPO training stage depicted in the right two figures shows more stable optimization patterns. Due to its characteristic of fine-tuning only on "hard tokens," it

maintains a relatively stable APR throughout the entire fine-tuning process. Moreover, we observe that the optimal checkpoints for the SFT and DPO stages can vary depending on the model size, emphasizing the need for adjustments based on validation set results. In summary, the ablation study validates the effectiveness of both SFT and DPO approaches and highlights the importance of selecting an appropriate training parameter to maximize adversarial attack performance.

## 5.4 Experiment on Generalizability Across Distribution Shifts

$RL^2EAK$ operates under the assumption that the attacker possesses prior knowledge about the domain of queries the victim is likely to propose. However, it is challenging to define the level of prior knowledge an attacker possess. To evaluate the robustness of our approach under varying degrees of adversarial knowledge, we conduct experiments across distinct data distribution scenarios. Specifically, we simulate four levels of attacker knowledge: (1) No prior knowledge; (2) High-level intra-domain knowledge, where the adversary identifies the general domain (e.g., Medical) but utilizes a dataset with a different distribution (e.g., training on PubMedQA (Jin et al., 2019) while the victim uses MedQA (Jin et al., 2021)); (3) Precise intra-domain knowledge, where the adversary possesses data exhibiting a distribution highly similar to the victim's; and (4) Misaligned knowledge, where the adversary anticipates a disparate domain (e.g., expecting Finance queries while the user submits Medical queries). All experiments utilize Qwen-2.5-3B-Instruct to ensure a fair comparison.

Table 2 details the performance across these settings. In the scenario representing a related but distinct distribution, transforming the training set from MedQA to PubMedQA while testing on MedQA, $RL^2EAK$ demonstrates significant robustness. Compared to the backbone baseline, our method reduces the APR from 41.99 to 30.27, achieving a 27.9% reduction. This indicates that $RL^2EAK$ maintains its effectiveness when the adversary utilizes a dataset from a similar domain that exhibits a distributional shift. In contrast, when the adversary utilizes a completely unrelated benchmark (e.g., training on FinanceBench while testing on PubMedQA), the APR improves by 53.8% compared with using backbone LLM. This phenomenon occurs because a model fine-tuned on a specific domain tends to generate queries or tokens intrinsic to that domain. Consequently, when the adversary relies on a different domain for training, the resulting domain mismatch leads to a significant deviation in the LLM's outputs, thereby undermining the attack's effectiveness.

Table 2: Attack performance with different levels of prior knowledge. "Benchmark" indicates the target testing dataset, "Prior Knowledge" indicates the dataset used for training.

| Benchmark | Prior Knowledge | APR ($\downarrow$) |
|---|---|---|
| MedQA | Base | 41.99 |
| | PubMedQA | 30.27 (27.9%$\downarrow$) |
| | FinanceBench | 51.90 (23.6 %$\uparrow$) |
| | MedQA | 21.60 (48.6 %$\downarrow$) |
| PubMedQA | Base | 628.50 |
| | MedQA | 484.42 (22.9%$\downarrow$) |
| | FinanceBench | 966.59 (53.8 %$\uparrow$) |
| | PubMedQA | 208.72 (66.8%$\downarrow$) |

## 6 Discussion

In this section, we discuss $RL^2EAK$'s compatibility with other attack vectors, potential mitigation approaches to $RL^2EAK$, and how $RL^2EAK$ can be repurposed as a proactive defense mechanism.

**$RL^2EAK$'s Compatibility.** We demonstrate $RL^2EAK$'s end-to-end prompt leakage attack pipeline with an order-based KV-cache side channel under the Longest Prefix Match (LPM) scheduling policy, yet $RL^2EAK$ is theoretically compatible with diverse scheduling mechanisms and inference engines. Specifically, since the query generation process is decoupled from the side-channel feedback, $RL^2EAK$ can be adapted to other environments by replacing the LPM-based token validator with a verifier tailored to the target mechanism. Regarding scheduling policies, beyond LPM, $RL^2EAK$ remains effective in First-Come-First-Served (FCFS) or priority-based scenarios by leveraging timing-based side channels (i.e., measuring Time-To-First-Token) to detect reuse (Zheng et al., 2024; Song et al., 2024). Regarding inference engines, our approach extends to generic frameworks like vLLM, which supports token-level KV cache matching and is consequently susceptible to these timing-based side channels. Moreover, $RL^2EAK$ applies to semantic cache scenarios widely adopted by frameworks such as GPTCache (GPTCache, 2025) and industrial providers (e.g., AWS (AWS, 2025), Microsoft (Microsoft, 2025b)). We consider designing effective and specialized validators for these alternative platforms a critical direction for future work, as it would further enhance the understanding of practical cache-based side-channel attacks.

**RL$^2$EAK's Potential Mitigation Approaches.** While RL$^2$EAK proposes a practical prompt leakage attack framework, we acknowledge that several approaches already exist to mitigate KV-cache-induced prompt leakage. First, multiple industrial applications (OpenAI, 2025a; DeepSeek, 2024) employ user-level cache isolation. This approach theoretically eliminates cache sharing between multi-tenant users entirely, effectively preventing side-channel attacks. Recently, Chu et al. (2025) proposed a selective KV-cache sharing mechanism to mitigate KV cache side-channel attacks through a more fine-grained permission management of cache sharing. However, it compromises the efficiency benefits of multi-tenant cache sharing. However, this approach directly prevents KV-cache sharing among multiple users. Under the specific domain service scenario defined by our threat model, where users' queries are highly correlated, isolating the KV-cache would significantly reduce the cache hit ratio, mitigating the design purpose of KV-cache. Besides the isolating-based approach, since RL$^2$EAK relies on response ordering changes caused by cache hits and employs a token-by-token attack strategy, adding timing noise to each response in the LLM service can disrupt response ordering and mitigate cache-based attacks (Zheng et al., 2024). However, this approach still requires balancing security-efficiency trade-offs, as excessive noise diminishes the practical efficiency advantages of cache sharing mechanisms. While these approaches can partially mitigate cache-based leakage, they all struggle with security-efficiency trade-offs. We think that RL$^2$EAK can serve as a risk assessment tool for privacy protection by evaluating vulnerability exposure in real deployment scenarios.

**Deploying RL$^2$EAK as a Cache-based Side Channel Risk Assessment Tool.** As discussed in this paper, RL$^2$EAK provides a practical prompt leakage attack framework. Beyond its adversarial capabilities, RL$^2$EAK can also serve as a risk assessment system for LLM serving mechanisms. In the above described multi-tenant service scenario, RL$^2$EAK can naturally serve as a direct KV cache risk assessment system by deploying a simulated attacker to evaluate each newly updated KV cache entry. The system performs attack simulations on queries to determine the minimum number of response queries required for potential query leakage, and correspondingly evicts the relevant KV cache entries before reaching this threshold.

Compared to other KV cache side channel mitigation approaches, which typically require efficiency and security trade-offs, RL$^2$EAK offers significant advantages. RL$^2$EAK can be deployed as a plugin on the same physical node without affecting the primary model's inference service. The computational overhead is negligible, as our experiments in Section 5.2 demonstrate that attack simulation can be effectively performed using lightweight models (e.g., 3B parameters), while the main service runs on much larger models, making the additional inference requirements computationally insignificant. Furthermore, the attack simulation operates without updating the main KV-cache storage, ensuring no interference with the primary inference pipeline.

Additionally, RL$^2$EAK's practicality can be further enhanced by leveraging service providers' prior knowledge. RL$^2$EAK's effectiveness relies on an auxiliary dataset, where dataset quality directly impacts framework effectiveness. As demonstrated in Section 5.4, the efficacy of RL$^2$EAK is heavily dependent on the distributional similarity between the target queries and the auxiliary knowledge. Service providers usually possess the most comprehensive prior knowledge about genuine user queries, theoretically enabling them to train more effective attacker simulators than real adversaries. RL$^2$EAK can thus identify high-risk cache entries before real adversaries can extract them, thereby preventing actual prompt leakage.

# 7 CONCLUSION

In this paper, we proposed RL$^2$EAK, a scalable and automated prompt leakage attack framework that utilizes SFT and auto-annotation DPO to enhance the adversary's capacity for inferring users' queries. RL$^2$EAK employed SFT as a cold-start process and leveraged the token-level guess accuracy from prompt leakage attacks as an implicit reward signal for automated DPO training. We evaluated RL$^2$EAK's effectiveness across three different model sizes on domain-specific datasets from medical and financial sectors. Experiments demonstrated that RL$^2$EAK significantly improved performance across all settings on both datasets, with attack efficiency improved by a maximum of 12.48× in terms of average requests required per token. Furthermore, we provided a comprehensive discussion of RL$^2$EAK's compatibility and scalability, along with its potential application as a risk assessment tool in real-world LLM serving systems.

ETHICS STATEMENT

Throughout our study, we have conducted a thorough ethical assessment of our research methodology and potential impacts. Our investigation does not involve human participants, personal information, or any sensitive data. The research relies exclusively on publicly available datasets for experimental evaluation, none of which contains any personally identifiable information. In recognition of the potential risks associated with prompt leakage techniques, we provide comprehensive discussions of mitigation approaches for the proposed framework in this paper. Furthermore, we emphasize that our proposed system can serve as a risk assessment tool to identify prompt leakage in time and protect user privacy in real-world applications.

REPRODUCIBILITY STATEMENT

We commit to open-sourcing our code and making it available once the paper is accepted. During the submission phase, the code will not be open-sourced to protect intellectual property. Upon acceptance, we will promptly release relevant code and resources under an appropriate open-source license. Specifically, our release will include the pre-processed datasets used in our experiments (MedQA and FinanceBench) and the main execution scripts for running our proposed framework. In the spirit of open science, we will make these artifacts publicly available to the extent possible without violating ethical, legal, or contractual obligations.

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

# A   ALGORITHM PROCEDURE OF RL²EAK

---

**Algorithm 1:** Operation pipeline of RL²EAK

---

**Input:** Auxiliary dataset $\mathcal{D}_{\text{aux}}$, base adversarial model $\pi_{\text{base}}$, hard token threshold $\gamma$, max guess attempts per token $\kappa$.

**Output:** Recovered target query $s^{\text{recover}}$.

```
/* Phase 1:  Local model fine-tuning                        */
```
1 Initialize preference dataset $\mathcal{D}_{\text{pref}} \leftarrow \emptyset$;
2 $\pi_{\text{SFT}} \leftarrow$ Supervised finetune $(\pi_{\text{base}}, \mathcal{D}_{\text{aux}})$;
3 **foreach** *Query $s \in \mathcal{D}_{aux}$* **do**
4     **foreach** *Token $s_t \in s$* **do**
5         **if** $\text{Rank}(s_t \mid s_{<t}) > \gamma$ ;                    `// Identify hard tokens`
6         **then**
7             $s^{\text{win}} \leftarrow [s_{<t}, s_t]$;                          `// Equation 2`
8             $s^{\text{lose}} \leftarrow$ Greedy Decode $(\pi_{\text{SFT}}, s_{<t})$ ;
9             $\mathcal{D}_{\text{pref}} \leftarrow \mathcal{D}_{\text{pref}} \cup \{(s^{\text{win}}, s^{\text{lose}})\}$;

10 $\pi_{\text{DPO}} \leftarrow$ DPO Train $(\pi_{\text{SFT}}, \mathcal{D}_{\text{pref}})$ ;                    `// Equation 4`

```
/* Phase 2:  Order-based KV-cache prompt leakage attack      */
```
11 Initialize $s^{\text{recover}} \leftarrow \emptyset$;
12 **while** *True* **do**
13     $\text{D}_{\text{hit}} \leftarrow 0$, $k \leftarrow 0$;
14     **while** $\text{D}_{hit} = 0$ *and* $k < \kappa$ **do**
15         $Q_{\text{gen}} \leftarrow$ Generate a batch of candidate tokens$(\pi_{\text{DPO}}, s^{\text{recover}})$;
16         $Q \leftarrow [Q_{\text{dummy}}, Q_{\text{gen}}, Q_{\text{dummy}}]$ ;                    `// Construct batch`
17         $\text{D}_{\text{hit}}, \tilde{q} \leftarrow$ Query server$(Q)$ ;                    `// Equation 5`
18         $k \leftarrow k + |Q_{\text{gen}}|$;
19     **if** $\text{D}_{hit} = 1$ **then**
20         Append token $\tilde{q}$ to $s^{\text{recover}}$;
21     **else**
22         **break** ;                            `// Stop if recovery fails`

23 **return** $s^{\text{recover}}$

---

# B   DECLARATION OF LLM USAGE

We used Large Language Models (LLMs) as writing assistance tools to polish language, improve clarity, and refine the presentation of our research findings. The LLMs did not contribute to research ideation, methodology design, experimental design, or the core technical contributions of this work. We take full responsibility for all content in this paper.

