# OpenReview forum: "RL$^2$eak: Reinforcement Learning Enhanced Prompt Leakage Attack in Multi-tenant Large Language Model Services"
_ICLR.cc/2026/Conference — Submitted to ICLR 2026_

### Official Review · Reviewer_RQv1 · 2025-10-17

**Soundness:** 2
**Presentation:** 1
**Contribution:** 2
**Rating:** 2
**Confidence:** 4

**Summary:**

This paper studies an active side-channel attack against shared LLM services, where the attacker tries to guess a victim user's query. This involves an attacker model, which predicts the possible next token in the victim prompt. The TTFT gap between generated queries provides an indicator of whether the target token is hit. To better enhance the prediction efficiency, this work proposes additionally training the attacker model on domain-specifc datasets. This leads to better effectiveness and efficiency.

**Strengths:**

- This paper discusses an interesting side channel attack about prompt leakage.

- It is intriguing to introduce the concept of RL to this attack scenario for a new opportunity.

**Weaknesses:**

- The discussion about the potential mitigation strategies is too simple.

    - An LLM inference engine keeping confidentiality in mind can easily mitigate this threat by only allowing the reuse of the KV cache in a user-isolated way.

    - What's more, the attack only targets the SGLang, and the method in Section 4.2 exploits the Longest Prefix Match scheduling policy specific to SGLang. This is an inappropriate setting. What about other scheduling policies, and what about other inference engines? It is necessary to discuss the design or setting in detail.

- Missing baseline comparison. As discussed in Related Works, there are many existing active side channel attacks. This work does not compare the proposed RL$^2$eak method to them, but claims to optimize the probing query generation process. This is unsubstantiated.

- Writing concerns

    - The motivation for introducing SFT and DPO to better align with the domain-specific dataset is unclear. I guess the intuition is that the target victim query that will be asked is similar to other queries. In this case, enabling the adversarial model to predict in a similar way will be helpful. If so, it is necessary to do decontamination between the training and the test set to avoid hacking issues.

    - What's more, why this attack is called an RL-enhanced attack is also unclear. Just because it operates in a trial-and-error way?

**Questions:**

- An algorithm presentation for this attack may facilitate understanding how the attack operates.

- It is unclear why introducing $m$ dummy queries is necessary (Section 4.2).

- What about extending to other inference engines, e.g., vLLM?

---

> ### Author Response · Authors · 2025-11-21
> **Author response - Part (1/3)**
>
> **Weakness 1: Discussion on potential mitigations and the choice of the target cache policy.**
>
> > The discussion about the potential mitigation strategies is too simple. An LLM inference engine keeping confidentiality in mind can easily mitigate this threat by only allowing the reuse of the KV cache in a user-isolated way. What's more, the attack only targets the SGLang, and the method in Section 4.2 exploits the Longest Prefix Match scheduling policy specific to SGLang. This is an inappropriate setting. What about other scheduling policies, and what about other inference engines? It is necessary to discuss the design or setting in detail.
>
> **Answer:** Thank you for your comments. We have addressed the discussion on attack methods under different scheduling schemes by updating the $\text{RL}^2\text{eak}$'s compatibility section **(Section 6)**.
>
> We clarify that while some frameworks propose using user-level isolation to prevent KV-cache side-channel attacks, this method inherently prevents KV-cache sharing among multiple users. In the specific domain service scenario defined in our threat model, where user queries are highly correlated, isolating the KV-cache significantly reduces the cache sharing ratio. Moreover, this assumption is also adopted by previous works [1,2].
>
> Regarding $\text{RL}^2\text{eak}$’s compatibility, we justify our choice on the caching mechanism (i.e., SGLang's Longest Prefix Match (LPM) mechanism) as follows:
>
> 1. **Other Scheduling Schemes:** Although SGLang currently supports LPM, our attack is not limited to this. In other scheduling settings, such as First-Come-First-Served (FCFS) and priority-based scenarios, we can still determine the length of reused tokens and execute a timing-based side-channel attack solely by measuring the Time-To-First-Token (TTFT) of queries [1-2]. Since the core focus of this paper is on the efficient generation of forged queries, not the exhaustive discussion of side-channel attacks in various real-world scenarios, we did not implement these variations.
> 2. **Compatibility with Other Engines:** The latest version of the vLLM framework [3] also supports token-level KV cache matching. Consequently, vLLM is also susceptible to timing-based side-channel attacks, which further demonstrates the broad applicability of the $\text{RL}^2\text{eak}$ framework.
>
> **Weakness 2: Justification on the choice of baselines.**
>
> > Missing baseline comparison. As discussed in Related Works, there are many existing active side channel attacks. This work does not compare the proposed RL2eak method to them, but claims to optimize the probing query generation process. This is unsubstantiated.
>
> **Answer: We clarify that in the KV-cache side-channel attack, the efficient generation of dummy queries and the verification using timing/order-based side channels are decoupled.** Therefore, in this work, we choose the order-based side channel as our baseline [4], because it is more efficient and stable. This corresponds to the "Base" column in our experimental results in Table 1, which uses a basic LLM model to generate dummy queries.
>
> We observe that the correct judgment rate for the order-based side channel is nearly 100% when a token match occurs. In contrast, for the TTFT (Time-to-First-Token) timing-based method, our preliminary experiments found that due to the inherent fluctuation in the server's statistics of query return times, relying solely on TTFT might lead to errors in restoring some tokens (although most tokens can still be restored), thus reducing the attack success rate.
>
> Furthermore, our main contribution lies in how to efficiently generate queries (query generation), rather than the side channel relied upon to determine whether the issued query token matches. Therefore, we believe it is unnecessary to compare against a baseline that only uses the TTFT-based side-channel attack.
>
> As for other side-channel attacks, such as those utilizing CPU/GPU hardware-level side channels, they are not compared as baselines in this study because they rely on **different threat models**.

---

> ### Author Response · Authors · 2025-11-21
> **Author response - Part (2/3)**
>
> **Weakness 3: Motivation on better improvement via SFT and DPO.**
>
> > The motivation for introducing SFT and DPO to better align with the domain-specific dataset is unclear. I guess the intuition is that the target victim query that will be asked is similar to other queries. In this case, enabling the adversarial model to predict in a similar way will be helpful. If so, it is necessary to do decontamination between the training and the test set to avoid hacking issues.
>
> **Answer:** We appreciate the reviewer’s comment regarding the motivation for SFT/DPO and the necessity of decontamination. We address the decontamination concern from **two perspectives**:
>
> 1. **Strict Data Decontamination (Standard Train-Test Split).** We clarify that our experimental setup strictly adheres to a standard training-testing split. We ensure there is no overlap between the samples used for SFT/DPO and the target victim queries in the test set.
> 2. **Robustness to Distributional Similarity (Cross-Dataset Generalization).** We have included new experiments (shown in the table below and Table 2 in the revised paper) to **investigate the impact of differing similarity in data distribution between the training and testing sets.** Specifically, we conducted experiments across four scenarios representing varying degrees of adversarial advantage: (1) No prior knowledge, (2) High-level intra-domain knowledge (e.g., training on PubMedQA, testing on MedQA), (3) Precise intra-domain knowledge, and (4) Misaligned knowledge (e.g., training on FinanceBench, testing on MedQA). As detailed in Table 2, our approach exhibits strong robustness, notably reducing the APR by 27.9% (from 41.99 to 30.27) in the high-level setting. In contrast, the misaligned knowledge scenario yields no improvement and actually increases the APR. The results demonstrate that even when the training and testing data exhibit distribution disparities, $\text{RL}^2\text{eak}$ remains robust in effectiveness.
>
> |  Benchmark   | Prior Knowledge |     APR (↓)      |
> | :----------: | :-------------: | :--------------: |
> |  **MedQA**   |      Base       |      41.99       |
> |              |    PubMedQA     | 30.27 (27.9% ↓)  |
> |              |  FinanceBench   | 51.90 (23.6% ↑)  |
> |              |      MedQA      | 21.60 (48.6% ↓)  |
> | **PubMedQA** |      Base       |      628.50      |
> |              |      MedQA      | 484.42 (22.9% ↓) |
> |              |  FinanceBench   | 966.59 (53.8% ↑) |
> |              |    PubMedQA     | 208.72 (66.8% ↓) |
>
> > **Table 1**: Attack performance with different levels of prior knowledge. "Benchmark" indicates the target testing dataset, "Prior Knowledge" indicates the dataset used for training the system.
>
> **Weakness 4: Clarification on “RL-enhanced” attack.**
>
> > What's more, why this attack is called an RL-enhanced attack is also unclear. Just because it operates in a trial-and-error way?
>
> **Answer:** We clarify that we refer to $\text{RL}^2\text{eak}$ as an RL-enhanced approach because its core implementation utilizes Direct Preference Optimization (DPO), which is recognized as a paradigm of offline Reinforcement Learning (RL) optimization methodology.
>
> **Question 1: Detailed algorithm presentation.**
>
> > An algorithm presentation for this attack may facilitate understanding how the attack operates.
>
> **Answer:** Thanks for your comment. We have added an algorithm in Appendix A in the revised paper to clearly demonstrate the operation pipeline of $\text{RL}^2\text{eak}$.

---

> ### Author Response · Authors · 2025-11-21
> **Author response - Part (3/3)**
>
> **Question 2: Necessity of dummy queries.**
>
> > It is unclear why introducing dummy queries is necessary (Section 4.2).
>
> **Answer:** We clarify that the reason for introducing "dummy queries" is that we need to get our candidate queries into the server queue to trigger the LPM mechanism. This operation makes the target query be processed earlier and returned to the dummy queries, therefore causing observable TTFT gaps from other candidate queries.
>
> The reason for proposing another $m$ dummy queries after candidate queries is: when the first dummy query is proposed and processed by the server, all dummy queries will receive the same token match count as the target match token. If these $m$ dummy queries are not added later, the target query, being proposed later, might be placed after all the dummy queries due to its original proposed order.
>
> The subsequent $m$ dummy queries proposed will cause all queries to form a "sandwich" structure, ensuring the target query is enclosed between the two parts of dummy queries. This design enhances the system's robustness.
>
> **Question 3: Generalization on inference engines.**
>
> > What about extending to other inference engines, e.g., vLLM?
>
> **Answer:** As discussed in Weaknesses 1 and 2, vLLM can also be attacked by a single variant of $\text{RL}^2\text{eak}$. This is achieved by simply replacing the token validator from an order-based mechanism check with a TTFT-based check.
>
> To make it clearer, we have updated the discussion in Section 6 regarding compatibility with various engine supports.
>
> [1] X. Zheng et al., “InputSnatch: Stealing Input in LLM Services via Timing Side-Channel Attacks,” arXiv:2411.18191.
>
> [2] L. Song et al., “The Early Bird Catches the Leak: Unveiling Timing Side Channels in LLM Serving Systems,” arXiv:2409.20002.
>
> [3] vLLM, “vLLM open-source repository”, https://github.com/vllm-project/vllm.
>
> [4] G. Wu et al., “I Know What You Asked: Prompt Leakage via KV-Cache Sharing in Multi-Tenant LLM Serving,” in NDSS 25, 2025.

---

> > ### Comment · Reviewer_RQv1 · 2025-11-23
> >
> > Thanks for the authors' responses. I have several concerns listed as follows.
> >
> > 1. The following claim does not make sense. If attackers leverage platform-specific features (e.g., the scheduling algorithm), this will limit the effectiveness of the proposed attack. We require additional empirical evidence to demonstrate extensibility to other scheduling algorithms and platforms, if the authors claim they are conquerable.
> > >  Since the core focus of this paper is on the efficient generation of forged queries, not the exhaustive discussion of side-channel attacks in various real-world scenarios, we did not implement these variations.
> >
> > 2. Concerning the Response to Weakness 4: Clarification on “RL-enhanced” attack, this technical concept is misused. Although DPO is inspired by RL theory, it is a supervised learning algorithm rather than an RL algorithm.

---

> ### Author Response · Authors · 2025-11-28
> **Further Author Response to Reviewer RQv1 - Part (1/1)**
>
> **Question 1:** **Discussion on the choice of the target cache policy.**
>
> > The following claim does not make sense. If attackers leverage platform-specific features (e.g., the scheduling algorithm), this will limit the effectiveness of the proposed attack. We require additional empirical evidence to demonstrate extensibility to other scheduling algorithms and platforms, if the authors claim they are conquerable.
>
> **Answer 1:** Thank you for your further comments. We have addressed the discussion on attack methods under different scheduling schemes and mitigations by updating the **Compatibility** section (Section 6).
>
> **Regarding Extensibility and Scheduling Policies:****
> **We clarify our choice of the target setting and address the extensibility of our framework as follows:
>
> 1. **Decoupled Framework Design:** We emphasize that our proposed framework features a **decoupled design**, separating the *Query Generation Module* from the *Token Validator* (the component responsible for verifying cache hits via side channels). The core contribution of this paper is the efficient generation of forged queries (dummy queries) that maximize cache reuse potential.
> 2. **Scope of the Empirical Study:** While our current implementation targets SGLang’s Longest Prefix Match (LPM) for validation, the attack logic remains applicable to other schedulers (e.g., FCFS or priority-based) by substituting the validator. For instance, in FCFS settings, a timing-based validator measuring Time-To-First-Token (TTFT) could be employed. However, designing and optimizing specific token validators for every possible scheduling algorithm requires distinct engineering efforts that are **out of the scope** of this paper. Our work focuses on the algorithmic efficiency of generating the attack queries themselves, rather than the implementation of side-channel measurement tools for multiple platforms.
> 3. **Future Work and Broad Applicability:** We have updated **Section 6** to explicitly position the exploration of specialized validator designs as a direction for future research. We illustrate that further advancements in validator design will complement the KV-cache side channel framework and continue to enhance the overall performance of the attack.
>
> **Question 2:**  **Clarification on “RL-enhanced” attack.**
>
> > Concerning the Response to Weakness 4: Clarification on “RL-enhanced” attack, this technical concept is misused. Although DPO is inspired by RL theory, it is a supervised learning algorithm rather than an RL algorithm.
>
> **Answer 2:** Thanks for the comments. We respectfully clarify that classifying DPO as a form of Reinforcement Learning is consistent with recent literature. As demonstrated in [1] and [2], DPO is widely recognized as an **Offline Reinforcement Learning (or offline preference-based RL)** algorithm because it mathematically optimizes the RL objective (maximizing reward under KL constraints), even though it is solved via a closed-form loss on static data.
>
> **However, we are happy to revise the terminology in the final version (e.g., to "DPO-enhanced") to avoid confusion.** We hope this clarification addresses your concern regarding the technical definition. If so, we kindly request your reconsideration of the rating.
>
> [1] A. Agarwal, C. Dann, and T. V. Marinov, “Design Considerations in Offline Preference-based RL,” In Proc. of ICML, 2025.
>
> [2] H. Ivison *et al.*, “Unpacking DPO and PPO: Disentangling Best Practices for Learning from Preference Feedback,” In Proc. of NuerIPS, 2024.

---

### Official Review · Reviewer_C8wA · 2025-10-27

**Soundness:** 2
**Presentation:** 2
**Contribution:** 2
**Rating:** 4
**Confidence:** 3

**Summary:**

This paper introduces RL^2eak, a framework to improve the efficiency of prompt leakage attacks that exploit shared KV-caches in multi-tenant LLMs. The method uses a two-stage fine-tuning process (SFT followed by DPO) and features a novel automated annotation technique for DPO that generates preference pairs from token generation difficulty. Experiments show that RL^2eak significantly reduces the number of queries required to reconstruct a victim's prompt.

**Strengths:**

1. The paper presents an interesting application of reinforcement learning techniques (SFT + DPO) to optimize cache-based side-channel attacks in LLM services.
2. The assumptions about adversary capabilities (domain knowledge but no specific prompt knowledge) are reasonable for real-world scenarios.

**Weaknesses:**

1. The paper's motivation is weak, as it fails to convincingly argue why reducing the number of attack requests is a significant problem.
2. Evaluations are limited to only two domain-specific datasets and exclusively use the Qwen-2.5 model family, without testing other mainstream open-source LLMs such as Llama, Deepseek, or GLM.

**Questions:**

1. The paper's central weakness is the lack of justification for why optimizing an already-feasible attack matters. Could the authors please elaborate on the significance of improving attack efficiency? In what specific, practical threat models is the number of queries the primary bottleneck for an adversary, to the extent that a 10x improvement moves the attack from being impractical to practical?
2. Experimental evaluations were conducted using only two domain datasets and the Qwen-2.5 model. How does the proposed method generalize to other knowledge-intensive domains beyond medical and financial contexts, such as legal documents, scientific literature, or software code? More importantly, does RL^2eak demonstrate consistent effectiveness when applied to other prominent open-source LLM families like Llama, Deepseek, or GLM models?

---

> ### Author Response · Authors · 2025-11-21
> **Author response - Part (1/3)**
>
> **Weakness 1/ question 1: Justification on motivation to improve attack efficiency.**
>
> > The paper's motivation is weak, as it fails to convincingly argue why reducing the number of attack requests is a significant problem. The paper's central weakness is the lack of justification for why optimizing an already-feasible attack matters. Could the authors please elaborate on the significance of improving attack efficiency? In what specific, practical threat models is the number of queries the primary bottleneck for an adversary, to the extent that a 10x improvement moves the attack from being impractical to practical?
>
> **Answer:** Thank you for your comment on the motivation of our work. We **have updated our description for motivation in Section 1** in the revised version. Our motivation for the reduction in the number of attack requests is supported by prior studies [1, 2]. These works establish that earlier KV cache side-channel attacks were hindered by a requirement for an impractically large number of requests to successfully extract information. Our paper addresses this by demonstrating that under more realistic assumptions, an attacker with certain domain knowledge can substantially lower the "average number of requests" (APR). A core objective of this paper is to alert the research community to the potentially underestimated risks of KV cache side-channel attacks when considering real-world attacker capabilities.
>
> **Weakness 2: Generalization on domains and LLM backbones.**
>
> > Experimental evaluations were conducted using only two domain datasets and the Qwen-2.5 model. How does the proposed method generalize to other knowledge-intensive domains beyond medical and financial contexts, such as legal documents, scientific literature, or software code? More importantly, does RL^2eak demonstrate consistent effectiveness when applied to other prominent open-source LLM families like Llama, Deepseek, or GLM models?
>
> **Answer:** Thank you for your insightful feedback. We have incorporated additional experiments to address your concerns regarding the generalization of our method.
>
> - Specifically, we have now included an evaluation on **a new medical dataset (PubMedQA)**, which further demonstrates the effectiveness and robustness of $\text{RL}^2\text{eak}$. across different domains. More details are listed in **Table A** below (Table 1 in Section 5.2 of the revised paper).
> - We expanded our testing to the **Llama 3.1 series of models,** and $\text{RL}^2\text{eak}$. consistently maintained its performance. The complete experimental results are presented **in Table B** below (Table 1 in Section 5.2 of the revised paper). The result illustrates the effectiveness of $\text{RL}^2\text{eak}$ under different model architectures.
> - We also conducted an additional experiment to evaluate the transformation performance of $\text{RL}^2\text{eak}$ in Section 5.4 of the revised paper. This experiment involved training on one dataset and testing on another dataset from the same domain. The results **in Table C** further validate the generalizability of $\text{RL}^2\text{eak}$ under different adversary assumptions. The results in Table B validate the generalizability of $\text{RL}^2\text{eak}$ under varying adversary assumptions.

---

> ### Author Response · Authors · 2025-11-21
> **Author response - Part (2/3)**
>
> ###
>
>
>
> |           Model           | Method | MedQA ASR_500 | ASR_1000 | ASR_10000 | APR | w/l |  FinanceBench ASR_500 | ASR_1000 | ASR_10000 | APR | w/l | PubMedQA ASR_500 | ASR_1000 | ASR_10000 | APR | w/l |
> | :-----------------------: | :----: | :------------------------: | :-------------------------: | :--------------------------: | :----------: | :----------: | :-------------------------------: | :--------------------------------: | :---------------------------------: | :-----------------: | :-----------------: | :---------------------------: | :----------------------------: | :-----------------------------: | :-------------: | :-------------: |
> | **Llama-3.1-8B-Instruct** |  Base  |            5.3%            |            27.3%            |            92.0%             |    60.00     |      -       |               0.0%                |                0.0%                |                60.0%                |       286.52        |          -          |             0.0%              |              0.0%              |              57.0%              |     801.71      |        -        |
> |                           |  SFT   |            8.7%            |            43.3%            |            98.0%             |    30.43     |    74.0%     |               62.0%               |             **86.0%**              |                98.0%                |        37.91        |      **98.0%**      |             13.0%             |           **33.0%**            |            **93.0%**            |     254.86      |    **94.0%**    |
> |                           |  DPO   |            8.0%            |            28.7%            |            92.7%             |    54.86     |    74.0%     |               0.0%                |                2.0%                |                66.0%                |       248.69        |        88.0%        |             0.0%              |              0.0%              |              60.0%              |     666.01      |      84.0%      |
> |                           | System |         **12.6%**          |          **48.7%**          |          **98.7%**           |  **28.31**   |  **79.3%**   |             **64.0%**             |             **86.0%**              |             **100.0%**              |      **33.89**      |      **98.0%**      |           **14.0%**           |           **33.0%**            |            **93.0%**            |   **241.48**    |    **94.0%**    |
>
> ***
> >**Table A**: Main experimental results on Llama-3.1-8B-Instruct model. DPO results are obtained by directly training the base model using System's auto-annotation approach.
> ###

---

> ### Author Response · Authors · 2025-11-21
> **Author response - Part (3/3)**
>
> |           Model           | Method | PubMedQA ASR_500 | ASR_1000 | ASR_10000 | APR | w/l |
> | :-----------------------: | :----: | :---------------------------: | :----------------------------: | :-----------------------------: | :-------------: | :-------------: |
> |   **Qwen-3B-Instruct**    |  Base  |             0.0%              |              3.0%              |              68.0%              |     628.50      |        -        |
> |                           |  SFT   |           **12.0%**           |             28.0%              |            **94.0%**            |     220.85      |    **91.0%**    |
> |                           |  DPO   |             1.0%              |              5.0%              |              71.0%              |     571.51      |      63.0%      |
> |                           | System |             11.0%             |           **31.0%**            |            **94.0%**            |   **208.72**    |    **91.0%**    |
> |   **Qwen-7B-Instruct**    |  Base  |             0.0%              |              1.0%              |              49.0%              |     943.05      |        -        |
> |                           |  SFT   |           **9.0%**            |             21.0%              |              86.0%              |     291.28      |      91.0%      |
> |                           |  DPO   |             0.0%              |              1.0%              |              49.0%              |     901.09      |      70.0%      |
> |                           | System |             6.0%              |           **22.0%**            |            **88.0%**            |   **274.91**    |    **91.0%**    |
> |   **Qwen-14B-Instruct**   |  Base  |             0.0%              |              0.0%              |              58.0%              |     796.08      |        -        |
> |                           |  SFT   |             4.0%              |             24.0%              |              81.0%              |     374.85      |      88.0%      |
> |                           |  DPO   |             0.0%              |              0.0%              |              59.0%              |     753.28      |      40.0%      |
> |                           | System |           **12.0%**           |           **30.0%**            |            **89.0%**            |   **304.76**    |    **88.0%**    |
> | **Llama-3.1-8B-Instruct** |  Base  |             0.0%              |              0.0%              |              57.0%              |     801.71      |        -        |
> |                           |  SFT   |             13.0%             |           **33.0%**            |            **93.0%**            |     254.86      |    **94.0%**    |
> |                           |  DPO   |             0.0%              |              0.0%              |              60.0%              |     666.01      |      84.0%      |
> |                           | System |           **14.0%**           |           **33.0%**            |            **93.0%**            |   **241.48**    |    **94.0%**    |
>
> > **Table B**: Main experimental results on PubMedQA datasets. DPO results are obtained by directly training the base model using System's auto-annotation approach.
>
> |  Benchmark   | Prior Knowledge |     APR (↓)      |
> | :----------: | :-------------: | :--------------: |
> |  **MedQA**   |      Base       |      41.99       |
> |              |    PubMedQA     | 30.27 (27.9% ↓)  |
> |              |  FinanceBench   | 51.90 (23.6% ↑)  |
> |              |      MedQA      | 21.60 (48.6% ↓)  |
> | **PubMedQA** |      Base       |      628.50      |
> |              |      MedQA      | 484.42 (22.9% ↓) |
> |              |  FinanceBench   | 966.59 (53.8% ↑) |
> |              |    PubMedQA     | 208.72 (66.8% ↓) |
>
> > **Table C**: Attack performance with different levels of prior knowledge. "Benchmark" indicates the target testing dataset, "Prior Knowledge" indicates the dataset used for training the system.
>
> [1] Z. Luo *et al.*, “Shadow in the Cache: Unveiling and Mitigating Privacy Risks of KV-cache in LLM Inference,” *arXiv*: arXiv:2508.09442.
>
> [2] C. Gu, X. L. Li, R. Kuditipudi, P. Liang, and T. Hashimoto, “Auditing Prompt Caching in Language Model APIs,” *arXiv*: arXiv:2502.07776.

---

> > ### Comment · Reviewer_C8wA · 2025-11-24
> >
> > Thank you for your response.
> > Could the manuscript include a comparison of attack completion times before and after optimization, similar to the analysis presented in Reference 1? Attack completion time is more intuitive than the number of requests for demonstrating the practical impact of your method. If the experimental results can show that the optimization significantly reduces attack duration—particularly by making previously impractical attacks feasible within reasonable timeframes—this would strengthen the contribution of your work. I would be happy to reconsider my rating based on such evidence.

---

> > > ### Author Response · Authors · 2025-11-28
> > > **Further Author Response to Reviewer C8wA - Part (3/3)**
> > >
> > > | PubMedQA / Model            | Method | ASR$_{500}$ | ASR$_{1000}$ | ASR$_{10000}$ | w/l       | ATRT      | APR        |
> > > | :------------------------ | :----- | :---------- | :----------- | :------------ | :-------- | :-------- | :--------- |
> > > | **Qwen-2.5-3B-Instruct**  | Base   | 0.0%        | 3.0%         | 68.0%         | -         | 22.06     | 628.50     |
> > > |                           | SFT    | **12.0%**   | 28.0%        | **94.0%**     | **91.0%** | 8.39      | 220.85     |
> > > |                           | DPO    | 1.0%        | 5.0%         | 71.0%         | 63.0%     | 19.70     | 571.51     |
> > > |                           | System | 11.0%       | **31.0%**    | **94.0%**     | **91.0%** | **7.68**  | **208.72** |
> > > | **Qwen-2.5-7B-Instruct**  | Base   | 0.0%        | 1.0%         | 49.0%         | -         | 31.66     | 943.05     |
> > > |                           | SFT    | **9.0%**    | 21.0%        | 86.0%         | 91.0%     | 10.93     | 291.28     |
> > > |                           | DPO    | 0.0%        | 1.0%         | 49.0%         | 70.0%     | 30.24     | 901.09     |
> > > |                           | System | 6.0%        | **22.0%**    | **88.0%**     | **91.0%** | **10.56** | **274.91** |
> > > | **Qwen-2.5-14B-Instruct** | Base   | 0.0%        | 0.0%         | 58.0%         | -         | 28.20     | 796.08     |
> > > |                           | SFT    | 4.0%        | 24.0%        | 81.0%         | 88.0%     | 12.17     | 374.85     |
> > > |                           | DPO    | 0.0%        | 0.0%         | 59.0%         | 40.0%     | 25.37     | 753.28     |
> > > |                           | System | **12.0%**   | **30.0%**    | **89.0%**     | **88.0%** | **11.21** | **304.76** |
> > > | **Llama-3.1-8B-Instruct** | Base   | 0.0%        | 0.0%         | 57.0%         | -         | 27.81     | 801.71     |
> > > |                           | SFT    | 13.0%       | **33.0%**    | **93.0%**     | **94.0%** | 8.81      | 254.86     |
> > > |                           | DPO    | 0.0%        | 0.0%         | 60.0%         | 84.0%     | 22.00     | 666.01     |
> > > |                           | System | **14.0%**   | **33.0%**    | **93.0%**     | **94.0%** | **8.47**  | **241.48** |
> > >
> > > > Table 3: Main experimental results on PubMed dataset. DPO results are obtained by directly training the base model using RL2 EAK’s auto-annotation approach.

---

> ### Author Response · Authors · 2025-11-28
> **Further Author Response to Reviewer C8wA - Part (1/3)**
>
> **Question 1: Concerns about completion time.**
>
> > Thank you for your response. Could the manuscript include a comparison of attack completion times before and after optimization, similar to the analysis presented in Reference 1? Attack completion time is more intuitive than the number of requests for demonstrating the practical impact of your method. If the experimental results can show that the optimization significantly reduces attack duration—particularly by making previously impractical attacks feasible within reasonable timeframes—this would strengthen the contribution of your work. I would be happy to reconsider my rating based on such evidence.
>
> **Answer 1:** Thank you for your insightful feedback. We agree that task completion time is an intuitive metric for directly demonstrating effectiveness. Accordingly, we have conducted additional experiments to incorporate this dimension in Section 5.2 of the latest version manuscript. Specifically, we introduced **the Average Token Recovery Time (ATRT)**, which measures the average time required to recover a single token, to the scenarios in Table 1. To ensure a fair comparison, all experiments utilized Qwen-2.5-3B-Instruct as the server model.
>
> The results are presented in the tables below (also updated in Table 1 of the revised manuscript). As shown in the table, on FinanceBench with Qwen-2.5-3B-Instruct, $\text{RL}^2\text{eak}$ reduces the ATRT from 9.92s to 1.59s, achieving a **6.24× speedup**. We find that the ATRT and APR metrics demonstrate a consistent trend. These results validate the practical contribution of $\text{RL}^2\text{eak}$. To illustrate the impact, consider a realistic scenario where a query contains 1,000 tokens. In such a case, $\text{RL}^2\text{eak}$ would reduce the task completion time from approximately 3 hours to less than 30 minutes. Furthermore, due to computational constraints, we currently deploy a relatively small LLM as the server to illustrate the relative improvement. With larger-scale LLMs serving as the server model, the response time per query batch would increase proportionally, making a larger absolute ATRT improvement brought by $\text{RL}^2\text{eak}$.

---

> ### Author Response · Authors · 2025-11-28
> **Further Author Response to Reviewer C8wA - Part (2/3)**
>
> | FinanceBench / Model | Method | ASR$_{500}$ | ASR$_{1000}$ | ASR$_{10000}$ | w/l | ATRT | APR |
> | :--- | :--- | :--- | :--- | :--- | :--- | :--- | :--- |
> | **Qwen-2.5-3B-Instruct** | Base | 2.0% | 2.0% | 54.0% | - | 9.92 | 240.81 |
> | | SFT | **88.0%** | **90.0%** | **100.0%** | **100.0%** | 1.61 | 20.81 |
> | | DPO | 2.0% | 2.0% | 56.0% | 68.0% | 9.90 | 238.81 |
> | | System | **88.0%** | **90.0%** | **100.0%** | **100.0%** | **1.59** | **19.29** |
> | **Qwen-2.5-7B-Instruct** | Base | 6.0% | 6.0% | 66.0% | - | 9.87 | 259.65 |
> | | SFT | 76.0% | 82.0% | **96.0%** | 94.0% | 2.32 | 69.63 |
> | | DPO | 6.0% | 6.0% | 72.0% | 84.0% | 8.94 | 246.38 |
> | | System | **86.0%** | **92.0%** | **96.0%** | **96.0%** | **2.19** | **66.87** |
> | **Qwen-2.5-14B-Instruct** | Base | 4.0% | 4.0% | 64.0% | - | 8.49 | 214.19 |
> | | SFT | 78.0% | **88.0%** | **98.0%** | 96.0% | 1.98 | 39.49 |
> | | DPO | 4.0% | 4.0% | 64.0% | 62.0% | 8.16 | 213.46 |
> | | System | **84.0%** | **88.0%** | **98.0%** | **98.0%** | **1.85** | **36.31** |
> | **Llama-3.1-8B-Instruct** | Base | 0.0% | 0.0% | 60.0% | - | 10.96 | 286.52 |
> | | SFT | 62.0% | **86.0%** | 98.0% | **98.0%** | 2.04 | 37.91 |
> | | DPO | 0.0% | 2.0% | 66.0% | 88.0% | 9.61 | 248.69 |
> | | System | **64.0%** | **86.0%** | **100.0%** | **98.0%** | **1.98** | **33.89** |
>
> > Table 1: Main experimental results on FinanceBench dataset. DPO results are obtained by directly training the base model using RL2 EAK’s auto-annotation approach.
>
> | MedQA / Model              | Method | ASR$_{500}$ | ASR$_{1000}$ | ASR$_{10000}$ | w/l       | ATRT     | APR       |
> | :------------------------ | :----- | :---------- | :----------- | :------------ | :-------- | :------- | :-------- |
> | **Qwen-2.5-3B-Instruct**  | Base   | 18.0%       | 37.3%        | 95.3%         | -         | 2.59     | 41.99     |
> |                           | SFT    | 27.3%       | 53.3%        | 98.0%         | 80.0%     | 2.20     | 26.37     |
> |                           | DPO    | 18.0%       | 38.0%        | 95.3%         | 48.7%     | 2.58     | 41.86     |
> |                           | System | **29.3%**   | **55.3%**    | **98.6%**     | **84.0%** | **2.12** | **21.60** |
> | **Qwen-2.5-7B-Instruct**  | Base   | 18.0%       | 36.0%        | 94.0%         | -         | 2.71     | 54.50     |
> |                           | SFT    | **24.0%**   | **49.3%**    | 94.7%         | 79.3%     | 2.47     | 44.50     |
> |                           | DPO    | 17.3%       | 37.3%        | 94.7%         | 55.3%     | 2.68     | 52.67     |
> |                           | System | 22.7%       | 48.0%        | **96.0%**     | **81.3%** | **2.35** | **40.49** |
> | **Qwen-2.5-14B-Instruct** | Base   | 23.3%       | 45.3%        | 94.7%         | -         | 2.47     | 44.35     |
> |                           | SFT    | **31.3%**   | 56.6%        | 96.0%         | 78.6%     | 2.14     | 33.69     |
> |                           | DPO    | 24.0%       | 46.0%        | 94.7%         | 68.0%     | 2.45     | 43.83     |
> |                           | System | 30.7%       | **60.0%**    | **97.3%**     | **82.0%** | **2.07** | **30.60** |
> | **Llama-3.1-8B-Instruct** | Base   | 5.3%        | 27.3%        | 92.0%         | -         | 3.16     | 60.00     |
> |                           | SFT    | 8.7%        | 43.3%        | 98.0%         | 74.0%     | 2.35     | 30.43     |
> |                           | DPO    | 8.0%        | 28.7%        | 92.7%         | 74.0%     | 3.02     | 54.86     |
> |                           | System | **12.6%**   | **48.7%**    | **98.7%**     | **79.3%** | **2.17** | **28.31** |
> > Table 2: Main experimental results on MedQA dataset. DPO results are obtained by directly training the base model using RL2 EAK’s auto-annotation approach.

---

### Official Review · Reviewer_sS1o · 2025-10-28

**Soundness:** 2
**Presentation:** 3
**Contribution:** 2
**Rating:** 4
**Confidence:** 3

**Summary:**

This paper explores the vulnerability of Key-Value cache sharing among users of multi-tenant serving frameworks. Building on prior works which have expanded attack surface of cache sharing mechanism, the authors propose RL2eak which involves domain knowledge of user queries by SFT and RL, in order to improve attack performance, mainly on efficiency. RL2eak first SFT base local model with domain-specific data, then leverage this SFT model to annotate training data for DPO, identifying "hard tokens" that are challenging to generate. Experiments demonstrate RL2eak improve attack performance across medical and finance scenarios, achieving a maximum 12.48\times reduction in average requests needed to guess one token.

**Strengths:**

1. **Good topic**: The authors choose a practical scenario of deployed LLMs where multi-tenant serving framework is used and Key-Value cache sharing mechanism is employed. And they explore the prompt leakage threat from shared caching mechanism, taking Time to Fist Token (TTFT) as side channel signal.
2. **Good intuition of main method**: the method is build on the intuition that the cached key-value pairs can be reused when multiple users submit prompts with identical prefixs (line 156), so an observable timing side-channel can effectively indicate the correspondence of dummy query and victim query. Moreover, involving more domain knowlegde to generate dummy queries is realistic and practical.

**Weaknesses:**

1. The main concern is the strong assumption that the attacker's queries and the victim's queries are on the same GPU node (line 196). But this is not always achievable because the scheduling mechanism of server is not accessible. Even if they are on the same GPU node, the GPU can process multiple batches of queries (not only the ones from victims and attackers), which means the response time can be influenced by other irrelative queries.
2. The authors indicate that only-SFT leads to poor attack performance primarily due to overfitting and mode collapse (line 248), but there is no demonstration of such limitation.
3. The test set and the train set are highly correlated since they are sampled from the same dataset (line 319-323), contributing to the good performance of RL2eak. However, in real-world scenarios, the victim queries and dummy queries can be less correlated. It is recommended to train RL2eak on dataset 1 and test on dataset 2 (dataset 1 and 2 are from the same domain but have different distribution) to observe whether the attack is practical.
4. The proposed method has limited contribution. The main difference to prior work is involving domain knowledge via RL when generating dummy queries. Moreover, in Table 1, RL2eak has not much improvement compared to SFT.

**Questions:**

1. What does the consistent prompt prefix "Help me to guess the input:" (line 356) mean? Is it the target prefix to steal from victim queries?
2. In Table 1, I don't see a 18.1% reduction of RL2eak in APR compared to SFT (line 375). Is that a mistake?
3. Although the authors use APR to quantify efficiency, what is the actual time cost to steal a prompt? And is it acceptable? The victim query may be terminated before all dummy queries are processed.

---

> ### Author Response · Authors · 2025-11-21
> **Author response - Part (1/3)**
>
> **Weakness 1: Assumption on the KV cache pool.**
>
> > The main concern is the strong assumption that the attacker's queries and the victim's queries are on the same GPU node (line 196). But this is not always achievable because the scheduling mechanism of server is not accessible. Even if they are on the same GPU node, the GPU can process multiple batches of queries (not only the ones from victims and attackers), which means the response time can be influenced by other irrelevant queries.
>
> **Answer:** We apologize for the lack of clarity that led to this misunderstanding. Our framework actually operates under the practical assumption that "the victim's query and the attacker's query are processed on a server node that shares the same KV cache pool." To the best of our knowledge, many large-scale industry companies (e.g., Microsoft [1], ByteDance [2]) currently utilize KV-cache pooling technology to persist the KV cache for an extended period. In such real-world scenarios, the assumption that the attacker and victim share the same KV cache pool is practical and common. Furthermore, previous works on KV-cache side-channels also share this as a consensus assumption. We have updated the relevant description **in Section 3.3**. Additionally, since our queries are processed in batches, we only need to consider the relative order of responses within the batch. This approach effectively mitigates the influence of irrelevant queries.
>
> **Weakness 2: Demonstration of overfitting of SFT.**
>
> > The authors indicate that only-SFT leads to poor attack performance primarily due to overfitting and mode collapse (line 248), but there is no demonstration of such limitation.
>
> **Answer:** Thanks for your comments. In fact, we have already verified this point in the experiments in this paper. In the ablation experiments in Section 5.3, we tested the ratio of guessing attempts at different training steps compared to the change ratio of APR in the baseline. It can be observed that the SFT process is extremely prone to model collapse as the training steps increase, with the number of guessing attempts becoming $3.70 \times$ the original requirement. We have updated the description in **Section 4.1** to avoid this misunderstanding.
>
> **Weakness 3：Evaluation on less correlated datasets.**
>
> > The test set and the train set are highly correlated since they are sampled from the same dataset (line 319-323), contributing to the good performance of RL2eak. However, in real-world scenarios, the victim queries and dummy queries can be less correlated. It is recommended to train RL2eak on dataset 1 and test on dataset 2 (dataset 1 and 2 are from the same domain but have different distribution) to observe whether the attack is practical.
>
> **Answer:** Thank you for this insightful suggestion. In response, we have expanded our experimental evaluation to assess the robustness of $\text{RL}^2\text{eak}$ across varying levels of adversarial prior knowledge, specifically targeting the data transfer scenario you described.
>
> We introduced four distinct settings to test generalizability: (1) **No prior knowledge**; (2) **High-level intra-domain knowledge**, where the adversary identifies the general domain (e.g., training on PubMedQA, testing on MedQA); (3) **Precise intra-domain knowledge**; and (4) **Misaligned knowledge**, where the adversary utilizes a disparate domain (e.g., training on FinanceBench, testing on MedQA).
>
> Under the **High-level intra-domain** setting, $\text{RL}^2\text{eak}$ demonstrates robustness, reducing the APR by 27.9% (lowering it from 41.99 to 30.27). Conversely, our experiments with the misaligned knowledge scenario show that training on a disparate domain (e.g., Finance) does not yield improvements, confirming that the attack relies on the model being biased toward tokens intrinsic to the target domain.
>
> These results confirm that $\text{RL}^2\text{eak}$ remains highly effective even when the training and testing data exhibit distribution disparities, provided they share the same general domain. We have included these results in the table below (Table 2 in the revised paper) and provided a detailed analysis in Section 5.4 of the revised paper.
>
> |  Benchmark   | Prior Knowledge |     APR (↓)      |
> | :----------: | :-------------: | :--------------: |
> |  **MedQA**   |      Base       |      41.99       |
> |              |    PubMedQA     | 30.27 (27.9% ↓)  |
> |              |  FinanceBench   | 51.90 (23.6% ↑)  |
> |              |      MedQA      | 21.60 (48.6% ↓)  |
> | **PubMedQA** |      Base       |      628.50      |
> |              |      MedQA      | 484.42 (22.9% ↓) |
> |              |  FinanceBench   | 966.59 (53.8% ↑) |
> |              |    PubMedQA     | 208.72 (66.8% ↓) |
>
> > **Table 1**: Attack performance with different levels of prior knowledge. "Benchmark" indicates the target testing dataset, "Prior Knowledge" indicates the dataset used for training the system.

---

> ### Author Response · Authors · 2025-11-21
> **Author response - Part (2/3)**
>
> **Weakness 4: Clarification on contributions.**
>
> > The proposed method has limited contribution. The main difference to prior work is involving domain knowledge via RL when generating dummy queries. Moreover, in Table 1, RL2eak has not much improvement compared to SFT.
>
> **Answer:** Thanks for your valuable comments. We clarify that the contribution of $\text{RL}^2\text{eak}$ is two-fold:
>
> 1. $\text{RL}^2\text{eak}$ is **the first work to propose that a side-channel attack can be optimized by model optimization.** Furthermore, its core contribution is utilizing an automated preference dataset construction approach to go beyond directly optimizing the model by a domain dataset via Supervised Fine-Tuning (SFT).
> 2. Regarding the comparison between DPO and SFT performance, we clarify the experimental findings. DPO consistently achieves an additional reduction of approximately 10-18% in APR across most scenarios, validating its effectiveness. The reason that the APR cannot be further significantly improved is that, to maintain the reasoning and logic capacity of the LLM for scalable scenarios, the APR cannot be reduced indefinitely. Therefore, when the APR is already low, further optimization is challenging. This phenomenon is more pronounced in knowledge-intensive domains.
>    Furthermore, **our results show that directly applying the DPO process to the base model yields only modest improvements.** This is because we primarily optimize for hard tokens, which provide weak insight into the domain-specific knowledge. However, when DPO is combined with SFT, the domain knowledge is learned through SFT, and the hard tokens then become the primary limiting factor for the Attack Probing Rate (APR). In this combined scenario, we validate DPO as a highly effective and robust solution compared to pure SFT.
>
> **Question 1: Meaning of the prompt prefix.**
>
> > What does the consistent prompt prefix "Help me to guess the input:" (line 356) mean? Is it the target prefix to steal from victim queries?
> **Answer:** No, "Help me to guess the input:" is not the target prefix we aim to steal from victim queries. This prefix serves two distinct purposes in our framework:
>
> 1. **Task Instruction for Red-Team LLM**: The prefix "Help me to guess the input:" is used to instruct the red-team LLM to understand and engage with the prompt reconstruction task. It helps the model familiarize itself with the objective of guessing user inputs.
> 2. **Bootstrap Token for Autoregressive Generation**: Due to the autoregressive characteristic of LLMs, a prefix is technically necessary to initiate the generation process. The model requires an initial prompt to condition on before it can begin generating the first token of the reconstructed query.
>
> Importantly, we do not assume knowledge of any specific prefix from the victim's actual queries. Our attack relies only on high-level domain knowledge (e.g., medical or financial domain) rather than any concrete prefix information from the target system. We have added clarification of the prompt prefix's role **in Section 5.2**.
>
> **Question 2: Clarification on the performance improvement.**
>
> > In Table 1, I don't see a 18.1% reduction of RL2eak in APR compared to SFT (line 375). Is that a mistake?
>
> **Answer:** We are sorry for the unclear description that led to this misunderstanding. We clarify that the value represents the relative decrease compared to the SFT-optimized result. In the MedQA evaluation of the Qwen-2.5-3B-Instruct model, the APR decreases from 26.37 to 21.60. The relative decrease is therefore calculated as $(26.37 - 21.60) / 26.37 \approx 0.1808$, which is approximately $18.1$%. We have updated the results description paragraph to provide **this clear explanation in Section 5.2**.

---

> ### Author Response · Authors · 2025-11-21
> **Author response - Part (3/3)**
>
> **Question 3: Justification on the time cost.**
>
> > Although the authors use APR to quantify efficiency, what is the actual time cost to steal a prompt? And is it acceptable? The victim query may be terminated before all dummy queries are processed.
>
> **Answer**: We appreciate your valuable comments. Our empirical experiments show that for a query of approximately 100 tokens, $\text{RL}^2\text{eak}$ requires around 5 minutes to recover the whole query. The required time depends on the model used by the server and attacker. However, it is important to note that this recovery time is significantly shorter than the theoretical duration that a victim's query remains in the cache, particularly under KV cache pooling techniques.
>
> The user's KV cache attack window is primarily determined by the total volume of other users' queries and KV storage, not whether the user's own query has been terminated. Even if the user's query ends, the attack remains feasible as long as its KV cache has not been evicted from the cache pool. Furthermore, as discussed in Weakness 1, many companies (e.g., Microsoft [1], ByteDance [2]) have adopted solutions to extend the KV cache storage time, which further enlarges the potential attack time window for $\text{RL}^2\text{eak}$. The cache persistence time can extend substantially longer due to pooling optimizations, providing a sufficiently large window for conducting attacks. Therefore, following previous works [3,4], we did not consider recovery time as a primary metric in our experiments but instead identified the user's Request Number (e.g., APR in the paper) as the main evaluation metric.
>
> [1] S. Feng *et al.*, “AdaptCache: KV Cache Native Storage Hierarchy for Low-Delay and High-Quality Language Model Serving,” in *ACM SOSP BigMem Workshop*, Oct. 2025.
>
> [2] ByteDance Inc., “InfiniStore open-source repository.” https://github.com/bytedance/InfiniStore.
>
> [3] L. Song et al., “The Early Bird Catches the Leak: Unveiling Timing Side Channels in LLM Serving Systems,” arXiv: arXiv:2409.20002.
>
> [4] G. Wu et al., “I Know What You Asked: Prompt Leakage via KV-Cache Sharing in Multi-Tenant LLM Serving,” in NDSS 25, 2025.

---

> ### Author Response · Authors · 2025-11-28
> **Further Author Response to Reviewer sS1o - Part (1/3)**
>
> Question: **Justification on the time cost.**
>
> > Although the authors use APR to quantify efficiency, what is the actual time cost to steal a prompt? And is it acceptable? The victim query may be terminated before all dummy queries are processed.
>
> In response to your concern regarding timing metrics, we have updated the evaluation in Section 5.2 of the latest revision of our paper. We believe these new results effectively address the issue. We believe the results can address your concern. Specifically, we have expanded our evaluation to include Average Token Recovery Time (ATRT) to illustrate realistic query recovery time. To ensure a fair comparison, all experiments were conducted using Qwen-2.5-3B-Instruct as the server model.
>
> >**Average Token Recovery Time (ATRT)** measures the average time required to recover a single token. This metric provides a direct and intuitive assessment of the practical improvement in efficiency when recovering user queries.
>
> The updated results, shown in the table below and in the revised manuscript, demonstrate significant efficiency gains. On FinanceBench, $\text{RL}^2\text{eak}$ lowers the ATRT from 9.92s to 1.59s, representing a **6.24× speedup**. These findings underscore the practical utility of our method. For instance, recovering a realistic 1,000-token system prompt would take approximately 3 hours with the baseline, whereas $\text{RL}^2\text{eak}$ completes the task in under 30 minutes. Furthermore, while we utilized a smaller LLM due to computational limits, the absolute time savings offered by $\text{RL}^2\text{eak}$ would be even more pronounced when targeting larger-scale server models with higher latency.
>
> **We eagerly await your feedback and are ready to respond to any further questions you may have.**

---

> ### Author Response · Authors · 2025-11-28
> **Further Author Response to Reviewer sS1o - Part (2/3)**
>
> | FinanceBench / Model | Method | ASR$_{500}$ | ASR$_{1000}$ | ASR$_{10000}$ | w/l | ATRT | APR |
> | :--- | :--- | :--- | :--- | :--- | :--- | :--- | :--- |
> | **Qwen-2.5-3B-Instruct** | Base | 2.0% | 2.0% | 54.0% | - | 9.92 | 240.81 |
> | | SFT | **88.0%** | **90.0%** | **100.0%** | **100.0%** | 1.61 | 20.81 |
> | | DPO | 2.0% | 2.0% | 56.0% | 68.0% | 9.90 | 238.81 |
> | | System | **88.0%** | **90.0%** | **100.0%** | **100.0%** | **1.59** | **19.29** |
> | **Qwen-2.5-7B-Instruct** | Base | 6.0% | 6.0% | 66.0% | - | 9.87 | 259.65 |
> | | SFT | 76.0% | 82.0% | **96.0%** | 94.0% | 2.32 | 69.63 |
> | | DPO | 6.0% | 6.0% | 72.0% | 84.0% | 8.94 | 246.38 |
> | | System | **86.0%** | **92.0%** | **96.0%** | **96.0%** | **2.19** | **66.87** |
> | **Qwen-2.5-14B-Instruct** | Base | 4.0% | 4.0% | 64.0% | - | 8.49 | 214.19 |
> | | SFT | 78.0% | **88.0%** | **98.0%** | 96.0% | 1.98 | 39.49 |
> | | DPO | 4.0% | 4.0% | 64.0% | 62.0% | 8.16 | 213.46 |
> | | System | **84.0%** | **88.0%** | **98.0%** | **98.0%** | **1.85** | **36.31** |
> | **Llama-3.1-8B-Instruct** | Base | 0.0% | 0.0% | 60.0% | - | 10.96 | 286.52 |
> | | SFT | 62.0% | **86.0%** | 98.0% | **98.0%** | 2.04 | 37.91 |
> | | DPO | 0.0% | 2.0% | 66.0% | 88.0% | 9.61 | 248.69 |
> | | System | **64.0%** | **86.0%** | **100.0%** | **98.0%** | **1.98** | **33.89** |
>
> > Table 1: Main experimental results on FinanceBench dataset. DPO results are obtained by directly training the base model using RL2 EAK’s auto-annotation approach.
>
> | MedQA / Model              | Method | ASR$_{500}$ | ASR$_{1000}$ | ASR$_{10000}$ | w/l       | ATRT     | APR       |
> | :------------------------ | :----- | :---------- | :----------- | :------------ | :-------- | :------- | :-------- |
> | **Qwen-2.5-3B-Instruct**  | Base   | 18.0%       | 37.3%        | 95.3%         | -         | 2.59     | 41.99     |
> |                           | SFT    | 27.3%       | 53.3%        | 98.0%         | 80.0%     | 2.20     | 26.37     |
> |                           | DPO    | 18.0%       | 38.0%        | 95.3%         | 48.7%     | 2.58     | 41.86     |
> |                           | System | **29.3%**   | **55.3%**    | **98.6%**     | **84.0%** | **2.12** | **21.60** |
> | **Qwen-2.5-7B-Instruct**  | Base   | 18.0%       | 36.0%        | 94.0%         | -         | 2.71     | 54.50     |
> |                           | SFT    | **24.0%**   | **49.3%**    | 94.7%         | 79.3%     | 2.47     | 44.50     |
> |                           | DPO    | 17.3%       | 37.3%        | 94.7%         | 55.3%     | 2.68     | 52.67     |
> |                           | System | 22.7%       | 48.0%        | **96.0%**     | **81.3%** | **2.35** | **40.49** |
> | **Qwen-2.5-14B-Instruct** | Base   | 23.3%       | 45.3%        | 94.7%         | -         | 2.47     | 44.35     |
> |                           | SFT    | **31.3%**   | 56.6%        | 96.0%         | 78.6%     | 2.14     | 33.69     |
> |                           | DPO    | 24.0%       | 46.0%        | 94.7%         | 68.0%     | 2.45     | 43.83     |
> |                           | System | 30.7%       | **60.0%**    | **97.3%**     | **82.0%** | **2.07** | **30.60** |
> | **Llama-3.1-8B-Instruct** | Base   | 5.3%        | 27.3%        | 92.0%         | -         | 3.16     | 60.00     |
> |                           | SFT    | 8.7%        | 43.3%        | 98.0%         | 74.0%     | 2.35     | 30.43     |
> |                           | DPO    | 8.0%        | 28.7%        | 92.7%         | 74.0%     | 3.02     | 54.86     |
> |                           | System | **12.6%**   | **48.7%**    | **98.7%**     | **79.3%** | **2.17** | **28.31** |
> > Table 2: Main experimental results on MedQA dataset. DPO results are obtained by directly training the base model using RL2 EAK’s auto-annotation approach.

---

> ### Author Response · Authors · 2025-11-28
> **Further Author Response to Reviewer sS1o - Part (3/3)**
>
> | PubMedQA / Model            | Method | ASR$_{500}$ | ASR$_{1000}$ | ASR$_{10000}$ | w/l       | ATRT      | APR        |
> | :------------------------ | :----- | :---------- | :----------- | :------------ | :-------- | :-------- | :--------- |
> | **Qwen-2.5-3B-Instruct**  | Base   | 0.0%        | 3.0%         | 68.0%         | -         | 22.06     | 628.50     |
> |                           | SFT    | **12.0%**   | 28.0%        | **94.0%**     | **91.0%** | 8.39      | 220.85     |
> |                           | DPO    | 1.0%        | 5.0%         | 71.0%         | 63.0%     | 19.70     | 571.51     |
> |                           | System | 11.0%       | **31.0%**    | **94.0%**     | **91.0%** | **7.68**  | **208.72** |
> | **Qwen-2.5-7B-Instruct**  | Base   | 0.0%        | 1.0%         | 49.0%         | -         | 31.66     | 943.05     |
> |                           | SFT    | **9.0%**    | 21.0%        | 86.0%         | 91.0%     | 10.93     | 291.28     |
> |                           | DPO    | 0.0%        | 1.0%         | 49.0%         | 70.0%     | 30.24     | 901.09     |
> |                           | System | 6.0%        | **22.0%**    | **88.0%**     | **91.0%** | **10.56** | **274.91** |
> | **Qwen-2.5-14B-Instruct** | Base   | 0.0%        | 0.0%         | 58.0%         | -         | 28.20     | 796.08     |
> |                           | SFT    | 4.0%        | 24.0%        | 81.0%         | 88.0%     | 12.17     | 374.85     |
> |                           | DPO    | 0.0%        | 0.0%         | 59.0%         | 40.0%     | 25.37     | 753.28     |
> |                           | System | **12.0%**   | **30.0%**    | **89.0%**     | **88.0%** | **11.21** | **304.76** |
> | **Llama-3.1-8B-Instruct** | Base   | 0.0%        | 0.0%         | 57.0%         | -         | 27.81     | 801.71     |
> |                           | SFT    | 13.0%       | **33.0%**    | **93.0%**     | **94.0%** | 8.81      | 254.86     |
> |                           | DPO    | 0.0%        | 0.0%         | 60.0%         | 84.0%     | 22.00     | 666.01     |
> |                           | System | **14.0%**   | **33.0%**    | **93.0%**     | **94.0%** | **8.47**  | **241.48** |
>
> > Table 3: Main experimental results on PubMed dataset. DPO results are obtained by directly training the base model using RL2 EAK’s auto-annotation approach.

---

### Official Review · Reviewer_zDf7 · 2025-10-30

**Soundness:** 3
**Presentation:** 3
**Contribution:** 3
**Rating:** 6
**Confidence:** 2

**Summary:**

This paper study an important safety problem when client uses LLM, where attacker might exploit KV-cache reuse in multi-tenant LLM services to induce prompt leakage. They propose RL@EAK, an attacker-model training pipeline using SFT then DPO that automatically constructs preference pairs by identifying "hard tokens", then optimizing a probe-generation policies to save probing queries. Experiments show large reduction in average request per token and improvement in attack success rate.

**Strengths:**

1. This is an important and novel topic for LLM safety, very worth studying.
2. The experiments are solid, evaluating realistic domains with meaningful metrics.
3. The attack optimization is solid and practical, author did provide ablation studies for each component of the pipeline.

**Weaknesses:**

1. It is assumed that the attacker knows the victim domain so they can construct the domain auxiliary dataset. This maybe optimistic in real deployments -- even the LLM is deployed on certain domain, the difference on subdomain between user's prompt and attackers' auxiliary dataset might harm the current red-teaming method.

2. Commercial LLM hosting stacks vary. It is assumed that the attacker knew the particular caching/scheduling behavior of LLMs. An analysis of robustness of the attack across different caching design might strengthen the paper.

**Questions:**

Please address the suggestions in weaknesses.

---

> ### Author Response · Authors · 2025-11-21
> **Author response**
>
> **Weakness 1: Assumption on high-level domain knowledge.**
>
> > It is assumed that the attacker knows the victim domain so they can construct the domain auxiliary dataset. This maybe optimistic in real deployments -- even the LLM is deployed on certain domain, the difference on subdomain between user's prompt and attackers' auxiliary dataset might harm the current red-teaming method.
>
> **Answer:** Thanks for your valuable comments. We have added an experiment and corresponding analysis in Section 5.4 to demonstrate $\text{RL}^2\text{eak}$'s effectiveness in data distribution disparity.
>
> Specifically, to evaluate the robustness of our approach under varying degrees of adversarial advantage, we conducted experiments across **four distinct scenarios**: (1) **No prior knowledge**; (2) **High-level intra-domain knowledge**, where the adversary identifies the general domain (e.g., training on PubMedQA, testing on MedQA); (3) **Precise intra-domain knowledge**; and (4) **Misaligned knowledge**, where the adversary utilizes a disparate domain (e.g., training on FinanceBench, testing on MedQA). Detailed results are shown in the table below (or you can find them in Table 2 of the revised paper).
>
> As presented in the Table, our approach exhibits strong robustness in the high-level intra-domain knowledge setting, successfully reducing the APR by 27.9% (from 41.99 to 30.27). In contrast, the misaligned knowledge experiment indicates that training on a disparate domain does not yield significant improvements and, in fact, causes the APR to increase. This outcome supports the conclusion drawn in our paper: domain-specific fine-tuning biases the model toward generating tokens intrinsic to that domain. Consequently, utilizing misaligned domain knowledge causes the model's outputs to deviate significantly, affecting the attack performance.
>
> | Benchmark | Prior Knowledge | APR (↓) |
> | :---: | :---: | :---: |
> | **MedQA** | Base | 41.99 |
> | | PubMedQA | 30.27 (27.9% ↓) |
> | | FinanceBench | 51.90 (23.6% ↑) |
> | | MedQA | 21.60 (48.6% ↓) |
> | **PubMedQA** | Base | 628.50 |
> | | MedQA | 484.42 (22.9% ↓) |
> | | FinanceBench | 966.59 (53.8% ↑) |
> | | PubMedQA | 208.72 (66.8% ↓) |
>
> > **Table 1**: Attack performance with different levels of prior knowledge. "Benchmark" indicates the target testing dataset, "Prior Knowledge" indicates the dataset used for training the system.
>
> **Weakness 2: Robustness against caching design.**
>
> > Commercial LLM hosting stacks vary. It is assumed that the attacker knew the particular caching/scheduling behavior of LLMs. An analysis of robustness of the attack across different caching design might strengthen the paper.
>
> **Answer:** Thanks for your comments. We have updated Section 6 to further discuss the $\text{RL}^2\text{eak}$’s compatibility and its applicability. As discussed in the paper, the core contribution of $\text{RL}^2\text{eak}$ lies in significantly accelerating the guessing of the user's query, thereby enhancing the attack's efficiency. In this process, the two main stages, guessing the user's input and using the side channel (i.e., different KV cache scheduling methods) to verify the correctness of the guessed token, are decoupled. Consequently, as long as the user's data is guessed effectively, $\text{RL}^2\text{eak}$ is essentially compatible with various verification frameworks that employ active cache-based side channels. For instance, regarding scheduling policies beyond LPM, $\text{RL}^2\text{eak}$ remains effective in First-Come-First-Served (FCFS) or priority-based scenarios by leveraging timing-based side channels (i.e., measuring Time-To-First-Token) [1,2] to detect cache reuse.
>
> [1] X. Zheng et al., “InputSnatch: Stealing Input in LLM Services via Timing Side-Channel Attacks,” arXiv:2411.18191.
>
> [2] L. Song et al., “The Early Bird Catches the Leak: Unveiling Timing Side Channels in LLM Serving Systems,” arXiv:2409.20002.

---

> > ### Comment · Reviewer_zDf7 · 2025-11-22
> >
> > Thank you for the detailed response. Please incorporate the revision in the final version.

---

> > > ### Author Response · Authors · 2025-11-28
> > > **Further Reply to Reviewer zDf7**
> > >
> > > We sincerely appreciate your kind and constructive comments. Your comments have significantly enhanced the quality of our manuscript. We will incorporate these revisions into the final version of the paper.

---

### Author Response · Authors · 2025-11-21
**Summary of rebuttal**

Dear PCs, SACs, ACs, Reviewers,

We sincerely thank the reviewers for their insightful and constructive feedback, which has helped us significantly improve the quality and clarity of our manuscript. We have carefully considered all comments and revised the paper accordingly. We have also submitted the revised manuscript, in which all modifications are highlighted in **blue**. We hope the following summary will assist the AC and future readers in better understanding our work and the rebuttal. We believe these revisions have strengthened the paper, and we provide our detailed point-by-point responses below.

**Addressing the Concerns Raised by Reviewer zDf7:**

- **Assumption on high-level domain knowledge:** We have added experiments in Section 5.4 to evaluate robustness under varying degrees of adversarial advantage: (1) No prior knowledge, (2) High-level intra-domain knowledge, (3) Precise intra-domain knowledge, and (4) Misaligned knowledge. The results demonstrate that $\text{RL}^2\text{eak}$ remains robust in the high-level intra-domain setting (reducing APR by 27.9%), while misaligned knowledge (e.g., training on Finance, testing on Medical) hampers performance, confirming our hypothesis on domain-specific bias.
- **Robustness against caching design:** We updated Section 6 to clarify that $\text{RL}^2\text{eak}$ decouples query guessing from verification. Consequently, our method is compatible with various scheduling policies (e.g., FCFS, priority-based) by leveraging timing-based side channels (Time-To-First-Token) alongside the Longest Prefix Match (LPM) mechanism.

**Addressing the Concerns Raised by Reviewer sS1o:**

- **Assumption on the KV cache pool:** We clarified in Section 3.3 that our assumption relies on the victim and attacker sharing a server node with a KV cache pool. We referenced real-world implementations, which utilize KV-cache pooling for persistence, validating the practicality of our threat model.
- **Demonstration of SFT overfitting:** We highlighted our ablation study (Section 5.3) which demonstrates that SFT is prone to model collapse as training steps increase, necessitating the DPO approach. We have updated the description in Section 4.1 to emphasize this finding.
- **Evaluation on less correlated datasets:** We conducted cross-dataset experiments (e.g., training on **PubMedQA**, testing on **MedQA**) to prove that $\text{RL}^2\text{eak}$ is effective even when training and testing data exhibit distribution disparities within the same general domain.
- **Time cost:** We clarify that our empirical experiments show that for a query of approximately 100 tokens, $\text{RL}^2\text{eak}$ requires around 5 minutes to recover the whole query. Notably, we clarify that the attack window is determined by **KV cache eviction** rather than the victim's query termination. Given that modern systems employ pooling techniques that extend cache persistence, supporting APR as our primary metric.

**Addressing the Concerns Raised by Reviewer** **C8wA****:**

- **Motivation for efficiency:** We updated Section 1 to emphasize that prior KV cache side-channel attacks were often impractical due to the prohibitive number of requests required, as documented in prior works. $\text{RL}^2\text{eak}$ bridges the gap between theoretical feasibility and practical exploitation.
- **Generalization on domains and models:** We expanded our evaluation to include the **PubMedQA** dataset and the **Llama-3.1** model series. Furthermore, we have conducted cross-dataset experiments to test $\text{RL}^2\text{eak}$’s generalizability under different attacker assumptions. The results confirm that $\text{RL}^2\text{eak}$ generalizes effectively across different attacker assumptions and model architectures.

**Addressing the Concerns Raised by Reviewer** **RQv1****:**

- **Mitigation and Target Cache Policy:** We discussed that while user isolation is a mitigation, it significantly reduces cache sharing efficiency. We also clarified that our attack is compatible with other engines, such as **vLLM** (which supports token-level matching), by adapting the verification mechanism.
- **Choice of Baselines:** We justified our use of order-based side channels as the baseline due to their stability compared to TTFT-based methods, while emphasizing that our main contribution is the efficient *generation* of queries, which is independent of the specific side channel used for verification.
- **SFT/DPO Motivation and Decontamination:** We clarify that we adhere to a strict train-test split. Furthermore, our new cross-dataset experiments demonstrate robustness to distributional shifts without data leakage.

Once again, we sincerely appreciate your time and valuable feedback. We hope that the revisions and clarifications we have provided effectively address the concerns and contribute to strengthening our submission.

Best,

The Authors

---

### Meta-Review · Area_Chair_Kkob · 2026-01-03

**Summary:**

While the problem setting is important and the paper contains substantial experimental work, the reviewers consistently raised concerns about novelty, threat-model realism, platform dependence, and conceptual framing, which ultimately outweigh the demonstrated empirical gains.

**Reviewer Concerns:**

Threat model realism and mitigation implications. Several reviewers questioned whether the assumed attacker capabilities (same cache pool, domain knowledge, timing observability) are representative of real deployments, and noted that straightforward mitigations (e.g., user-isolated caching) would largely defeat the attack. The rebuttal discusses these points but does not convincingly demonstrate that RLeak meaningfully changes the defender–attacker tradeoff.

Limited novelty and conceptual contribution. Multiple reviewers noted that the main contribution amounts to applying existing techniques (SFT + DPO) to optimize an already-known side-channel attack. While the empirical improvements are non-trivial, the work does not introduce new learning principles, new attack surfaces, or fundamentally new security insights beyond efficiency tuning.

**Reviewer Scores:**

sS1o and C8wA will remain negative given that the realistic of the attack is not addressed.

---

### Decision · Program_Chairs · 2026-01-26

Reject